# MATHCONSTRUCT: Challenging LLM Reasoning with Constructive Proofs

Mislav Balunović* [1 2]   Jasper Dekoninck* [1]   Nikola Jovanović [1]   Ivo Petrov [2]   Martin Vechev [1]

## Abstract

While Large Language Models (LLMs) demonstrate impressive performance in mathematics, existing math benchmarks come with significant limitations. Many focus on problems with fixed ground-truth answers, and are often saturated due to problem simplicity or the viability of guessing or memorization. Crucially, they capture only a narrow subset of relevant math problems. To address this research gap, we introduce MATHCONSTRUCT, a new benchmark of 126 challenging problems sourced from various math competitions, which targets *constructive proofs*, a widely encountered problem type requiring the construction of mathematical objects with specific properties. These proofs are particularly suitable for LLM evaluation, as solution correctness can be easily verified. Our automated verifiers also enable MATHCONSTRUCT to generate problem variations, used to evaluate robustness. State-of-the-art LLMs solve only 54% of MATHCONSTRUCT problems, highlighting its complexity and importance for LLM evaluation.

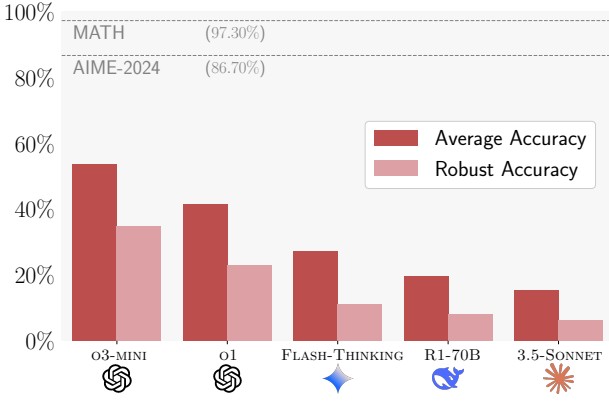

Figure 1: Accuracy of LLMs on MATHCONSTRUCT, highlighting the difficulty of constructive proofs. We compute Robust Accuracy by requiring the model to consistently solve a problem across a set of problem variations.

## 1. Introduction

Evaluating the mathematical reasoning abilities of Large Language Models (LLMs) requires high-quality public benchmarks that accurately measure progress. As shown in Fig. 1, existing benchmarks, such as MATH (Hendrycks et al., 2021), are becoming increasingly saturated as state-of-the-art models improve, highlighting the need for more challenging evaluation tasks. Many complex mathematical problems involve *proofs*, which are a fundamental component of advanced reasoning. However, current benchmarks primarily focus on problems where LLM outputs can be directly compared to ground truth answers, making them unsuitable for evaluating proofs. A promising alternative,

formalized proof generation, requires LLMs to generate proofs that can be verified by automated theorem provers such as Lean (de Moura & Ullrich, 2021). Unfortunately, even models explicitly fine-tuned for this task struggle to perform well (Xin et al., 2024). Furthermore, this approach does not fully leverage the strong natural language reasoning capabilities of LLMs. This raises an important question: Is there a class of proof-based problems that are both challenging for LLMs and easy to verify for correctness?

**Constructive proofs**   One important class of proofs, commonly appearing in real-world applications and advanced mathematical competitions, involves *constructive proofs*. These proofs establish a mathematical result by explicitly constructing an object—such as a set, matrix, or graph—that satisfies specific constraints.

For instance, one of the most challenging problems from the 2022 International Mathematical Olympiad (IMO), shown in Fig. 2 (left), requires constructing an $n \times n$ matrix that maximizes a particular quantity. More generally, disproving a conjecture often involves constructing a counterexample, as seen in Cantor's diagonal argument (Cantor, 1890). Similarly, proving a bound frequently requires constructing an object that achieves that bound, as in the proof of the Four Color Theorem (Appel & Haken, 1989).

*Equal contribution [1]Department of Computer Science, ETH Zurich [2]INSAIT, Sofia University "St. Kliment Ohridski". Correspondence to: Mislav Balunović <mislav.balunovic@inf.ethz.ch>.

*Proceedings of the 42nd International Conference on Machine Learning*, Vancouver, Canada. PMLR 267, 2025. Copyright 2025 by the author(s).

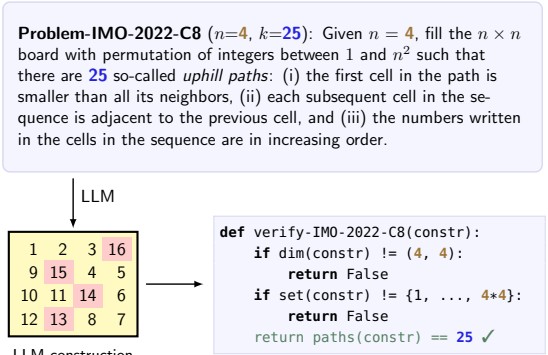
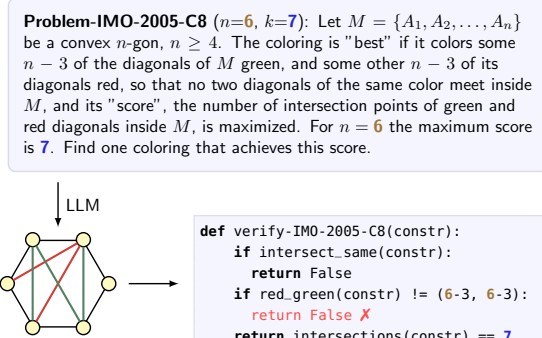

Figure 2: Two sample problems from MATHCONSTRUCT, each consisting of a natural language statement and a verifier function that returns a boolean value indicating the validity of a proposed construction. The ability to easily generate problem variations (*values colored in blue and brown*), the complexity of the required constructions, and the difficulty of the problems make MATHCONSTRUCT well-suited for evaluating LLMs' reasoning abilities.

Constructive proofs are particularly well-suited for LLM benchmarking as coming up with valid constructions is often difficult for humans, and thus likely to also challenge the models. Yet, verifying if a proposed construction satisfies the problem constraints is usually straightforward, enabling the use of automated verifiers to judge model responses.

**Our benchmark: MATHCONSTRUCT**  Leveraging this, we introduce MATHCONSTRUCT, a new benchmark designed to evaluate LLMs' reasoning capabilities through constructive proofs. MATHCONSTRUCT consists of 126 unique problems sourced from olympiad-level mathematics competitions. Each problem is encoded as a natural language statement and a corresponding verifier function that determines the correctness of a proposed construction (see Sec. 3 for details). In Fig. 2, we illustrate two sample problems from MATHCONSTRUCT.

Beyond its challenging nature and ease of verification, MATHCONSTRUCT has several features that make it particularly valuable for LLM evaluation. Most importantly, all problem statements are phrased symbolically, enabling systematic generation of *variations* that test models' robustness to small changes in problem parameters. Second, the required constructions often involve complex mathematical objects (e.g., matrices, colorings) rather than simple numerical answers, making guesswork and memorization less effective. Finally, our rigorous hand-curation process (detailed in Sec. 3) ensures high problem quality, robustness against brute-force solutions, and broad coverage across mathematical domains.

**Evaluation**  In Sec. 4, we evaluate state-of-the-art LLMs on MATHCONSTRUCT, including GPT-4O and O3-MINI (Jaech et al., 2024), the GEMINI family (Reid et al., 2024), and the CLAUDE family (Anthropic, 2024).

By generating variations of each problem, we evaluate the models on 475 distinct problem instances. Even with access to code execution, these models struggle with MATHCONSTRUCT. The best model achieves only 54% accuracy, as shown in Fig. 1; complete results are presented in Table 2. We further provide a thorough analysis of LLMs' failure modes, and study the impact of different variations, problem brute-forceability, and data contamination on performance. Our code is available at https://github.com/eth-sri/mathconstruct.

**Contributions**  Our key contributions are:

- We propose MATHCONSTRUCT, a benchmark of 126 challenging constructive problems (Sec. 3).
- We conduct a rigorous evaluation of 14 state-of-the-art LLMs on MATHCONSTRUCT, demonstrating its difficulty and significance for LLM evaluation (Sec. 4.1).
- We provide a detailed analysis of LLM performance on MATHCONSTRUCT, including the impact of various factors on model performance (Secs. 4.2–4.5).

## 2. Related Work

This section reviews related work on mathematical benchmarking and constructive proofs in machine learning.

**Easier math benchmarks**  Most math benchmarks for LLMs focus on problems where the final answer is a numerical value or algebraic expression that can be compared with a fixed ground truth. Among these, early benchmarks such as GSM8K (Cobbe et al., 2021) and MATH (Hendrycks et al., 2021) have been saturated by recent models (Jaech et al., 2024; DeepSeek-AI, 2025). More advanced problem sets, such as AIME 2024[1], are more difficult, yet state-of-the-art models still solve ≈87%.

---

[1]See American Invitational Mathematics Examination

**Olympiad-based benchmarks** To introduce more complexity, newer benchmarks such as OlympiadBench (He et al., 2024), Omni-MATH (Gao et al., 2024), and HARP (Yue et al., 2024) incorporate olympiad-level problems, including image-based and multilingual problems. However, these benchmarks still rely on fixed-answer verification, often using an LLM as a judge for comparison. In contrast, MATHCONSTRUCT focuses on constructive proofs rather than verifying predefined answers, providing a more challenging evaluation of reasoning abilities. This difference in evaluation methodology is highlighted in a problem statement that appears both in Omni-Math and MATHCONSTRUCT. Specifically, in the problem on the left of Fig. 2, Omni-MATH considers a lower bound, $2n(n-1)+1$, as the correct answer, whereas MATHCONSTRUCT requires models to construct an object that achieves the lower bound, which is significantly more challenging in this case.

**Private benchmarks** Several private benchmarks, such as FrontierMath (Glazer et al., 2024) and LastExam (Phan et al., 2025), have recently been introduced. Although FrontierMath includes problems that require more complex verification methods, the private nature of these benchmarks prevents large-scale use and makes it difficult to evaluate the progress of the field as a whole.

**Benchmarks with variations** None of the above benchmarks incorporate problem variations, even though their value has been acknowledged in prior work. In particular, recent efforts in related domains, such as GSM-Symbolic (Mirzadeh et al., 2024), Putnam-Axiom (Gulati et al., 2024), and UTMath (Yang et al., 2024) use symbolic problem reformulation. MATHCONSTRUCT aims to extend this line of work by applying variations to problems that require the construction of complex mathematical objects.

**Formal math benchmarks** Another category of benchmarks evaluates formal theorem proving, requiring solutions in languages such as Lean (de Moura & Ullrich, 2021). Examples include miniF2F (Zheng et al., 2022), FIMO (Liu et al., 2023), and PutnamBench (Tsoukalas et al., 2024). Despite specialized training (Xin et al., 2024), LLMs struggle to solve more than a few problems in this environment, indicating significant room for improvement.

**Logical reasoning benchmarks** Orthogonally, a large body of work evaluates logical reasoning in LLMs on logical puzzles and satisfiability problems (Gui et al., 2024; Mittal et al., 2024; Ma et al., 2024), many of which are NP-Hard (Fan et al., 2024). Examples include puzzles such as the knapsack problem and Sudoku. These tasks primarily test algorithmic problem-solving rather than mathematical reasoning. Furthermore, the problems are often well-known,

making them susceptible to memorization. While most problems in MATHCONSTRUCT are also NP-Hard, they ensure tractability for human solvers using pen and paper, as they originate from real mathematics competitions. This makes MATHCONSTRUCT a more direct evaluation of reasoning ability in mathematical contexts.

**Constructions with machine learning** Machine learning has also been previously applied to mathematical object construction and pattern discovery. Wagner (2021) use reinforcement learning to find counterexamples that disprove conjectures in graph theory. Other works leverage neural networks for combinatorial optimization (Bello et al., 2017; Gasse et al., 2019). More broadly, machine learning has been used to identify relationships between mathematical objects, as seen in Davies et al. (2021) and Davila (2024).

## 3. MATHCONSTRUCT

Constructive proofs are a powerful tool for mathematicians, but turning these into a benchmark for evaluating the reasoning abilities of LLMs requires significant effort. In this section, we describe our approach for the creation of a reliable benchmark of difficult constructive proof problems, detailing the key steps: problem selection (Sec. 3.1), problem encoding (Sec. 3.2), and post-hoc problem review (Sec. 3.3). Further details on the development process are given in App. A.

### 3.1. Problem Selection

As a first step in the creation of MATHCONSTRUCT, our team consisting of students with significant experience with math competitions preselected a preliminary set of constructive proof problems. To ensure quality, the problems were exclusively sourced from reputable high-level mathematics competitions, including high-school olympiads, undergraduate contests, and national high-school competitions. In total, our team read around 3500 problems from archives of 20 different competitions, selecting 158 possible problems that met our relevance and quality criteria.

**Problem selection criteria** Our criteria were as follows:

- *Difficult construction*: For MATHCONSTRUCT to be future-proof and pose a challenge to current and future models, obtaining the required construction should involve non-trivial reasoning, and constitute the majority of the official solution to the competition problem.

- *Complex objects*: To further ensure difficulty and reduce the probability of lucky guessing the answer, the required object should be non-trivial, i.e., the space of possible constructions should be large. In particular, we generally avoid problems where the result of the construction is a single integer.

```
# Symbolic problem statement P
problem = "Find an n × n matrix of
rank ≤ 3 with zeros on the main diagonal
and positive reals off the diagonal."

# Concrete parameters θ
parameters = {"n": 10}

# Verification function Vθ
def verify(construction):
    if dim(construction) != (n, n):
        return 0, "Wrong dimensions"
    if any(construction[i][i] != 0):
        return 0, "Non-zero diagonal"
    if any(construction[i][j] <= 0):
        return 0, "Off-diagonal <= 0"
    if matrix_rank(construction) > 3:
        return 0, "Rank > 3"
    return 1, "Correct"

# Variation generator
def generate_variation():
    return {"n": random.randint(6, 15)}

# Formatting instructions
format = "Present your construction as a
matrix in LaTeX enclosed in \boxed{}"
```

Figure 3: A problem encoding consisting of a symbolic problem statement $P$, concrete parameters $\theta$, and a verification function $V_\theta$. We also implement a variation generator, and provide formatting instructions for the construction.

- *Tractable verification*: On the other hand, the verification function must be straightforward to implement relative to the problem difficulty, and feasible to run.

- *Variations*: To distinguish genuine reasoning from memorization or lucky guessing, we prioritize problems that allow for multiple variations by altering parameters. In Sec. 4 we introduce *robust accuracy*, requiring models to solve multiple variations to be considered successful on a given problem. Since a human who knows the general solution can solve each variation trivially, this criterion ensures that the model has a similar level of understanding.

**Adapting different problem types**   The most common problems in MATHCONSTRUCT are *Any*-problems, which ask for any object satisfying the given constraints. However, several other problem types can be adapted to fit our criteria. For example, *Inf*-problems originally ask for a proof that there are infinitely many objects that satisfy the given constraints—these often require coming up with an infinite solution class, whose knowledge can be demonstrated by producing $k$ fitting constructions for $k \to \infty$. Similar reasoning holds for *All*-problems, which ask for all

objects satisfying some constraints. Finally, *Max*-problems originally ask for the maximum (or minimum) value of a function of an object. Often, these are solved by first proving a lower (resp. upper) bound and then constructing an object that achieves this bound. If the second part is sufficiently difficult, these problems are good candidates for MATHCONSTRUCT. Importantly, when adapting problems to become constructive proofs, we confirm that the quality and difficulty criteria above hold, *even after the necessary problem modifications*.

### 3.2. Math Construction Problems

We now discuss the formalization of construction problems and their encoding into a format that can be used for programmatic evaluation.

**Definition 3.1** (Construction Problem). A construction problem is a tuple $(P, \theta, V_\theta)$ where $P$ is a symbolic problem statement in natural language, $\theta$ are concrete parameters that replace symbolic variables in the problem statement, and $V_\theta \colon \mathcal{O} \to \{0, 1\}$ is a verification function that takes an object and checks whether it satisfies the constraints of the problem. A valid solution or a *constructive proof* for this problem is any object $o \in \mathcal{O}$ such that $V_\theta(o) = 1$.

Each problem is encoded as a single Python file, containing all of its components $P, \theta,$ and $V_\theta$. Additionally, the file includes a variation generator function which generates a set of variations of the problem by plugging in different values for the parameters $\theta$, and formatting instructions that instruct the model to output the solution in a specific format.

An example of an encoded MATHCONSTRUCT problem is given in Fig. 3. Here, the problem statement is $P =$ "*Find an $n \times n$ matrix...*", the parameters are $\theta = \{n : 10\}$, and the verification function is given by

$$V_\theta(M) = \operatorname{rank}(M) \le 3 \wedge M_{i,i} = 0 \wedge (i = j \vee M_{i,j} > 0).$$

When solving the problem, the model is given a concretized problem statement obtained by replacing the parameters in the problem statement $P$ with concrete parameter values $\theta$. Similar to human contestants, the model does not have access to the verification function $V_\theta$.

We now further discuss each of the problem components.

**Symbolic problem statement**   While many problems are originally stated with concrete values, they can often be generalized to obtain a *symbolic problem statement $P$*. We do this for every problem where it is possible, replacing concrete values with symbolic parameters $\theta$, allowing us to plug in different values for the parameter. For example, any positive integer can replace parameter $n$ in the problem statement in Fig. 3.

**Verification**    In our implementation, the verification function $V_\theta$ takes a construction and returns a tuple $(c, f)$ where $c$ is a boolean indicating whether the construction satisfies the constraints of the problem, and $f$ is a detailed feedback string that when $c = \text{false}$ explains why the construction is incorrect. For example, in the problem in Fig. 3, the verification function checks that the matrix has the correct dimensions, the diagonal is zero, the off-diagonal elements are positive, and the rank is at most 3. We use the feedback strings during our review process for manual quality checks (see Sec. 3.3). Following the criteria from Sec. 3.1, we see that despite the significant difficulty of the problem in Fig. 3, the verification function is simple to implement and run, and the feedback is straightforward to understand.

**Variations**    Given a problem, humans can typically find a general solution that works for all parameter values. For example, to solve the problem in Fig. 3 (and the majority of our problems), one can find a general procedure that works for any $n$. Most popular benchmarks (Hendrycks et al., 2021; Gao et al., 2024; He et al., 2024) do not have symbolic variations, meaning it is difficult to evaluate whether the model can generalize to similar problems or simply guessed the answer. Instead, we create a number of variations for each problem, allowing us to study the robustness of the model to changes in the problem parameters.

**Definition 3.2** (Problem Variations). Problem variations are a set of construction problems that share the same symbolic problem statement $P$ and verification function $V_\theta$, but have different problem parameters $\theta_1, \theta_2, \ldots, \theta_k$.

**Formatting and parsing**    Since the mathematical objects involved in constructive proofs have a wide variety of types, we also provide formatting instructions for each problem which can be used to prompt the model to output the solution in a specific format. Furthermore, we create a specialized parser, detailed in App. B, that can parse arbitrary combinations of lists, matrices, and LaTeX objects from natural language solutions. By default, models evaluated on our benchmark receive detailed feedback from our parser, allowing them to correct their solutions over multiple rounds, which reduces the risk of syntax errors.

### 3.3. Problem Review

We performed a review of MATHCONSTRUCT problems, in each stage discarding problems that did not meet our quality criteria and revising those that could be improved.

**Manual quality checks**    First, each problem author was tasked with implementing a generic solution function that computes valid constructions for the problem. Additionally, extensive unit tests were implemented to check each problem's verification and solution functions, ensuring correct

Table 1: Summary of MATHCONSTRUCT constructive proof problems by their category and type.

| Category \ Type | Any- | All- | Inf- | Max- | Σ |
|---|---|---|---|---|---|
| Combinatorics | 19 | 6 | 1 | 18 | 44 |
| Algebra | 11 | 5 | 1 | 7 | 24 |
| Number Theory | 26 | 7 | 15 | 6 | 54 |
| Geometry | 4 | 0 | 0 | 0 | 4 |
| Σ | 61 | 18 | 17 | 31 | 126 |

implementation. Next, each author was asked to ensure that their problem is solvable by a human using pen and paper, to ensure that we are testing reasoning, and not merely calculation skills. We remark that some of our problems may include more calculation than typical in human competitions (e.g., writing the complete $n \times n$ matrix instead of simply describing it)—however, this step is trivial once the correct insight for the problem is found. Finally, in a peer review process, each problem was reviewed by at least one other team member, checking that the problem statement is sound and clear, that the verification function is feasible and returns informative feedback in case of errors, and that the problem is sufficiently challenging and of high quality.

**Automated quality checks**    Complementing manual review, we also implemented automated checks. First, we verified that all problems are solvable by an LLM when explicitly given the solution. This verifies that the formatting instructions are unambiguous and that our parser and verifier are error-free. Second, we flagged for additional review all problems where our solution is longer than 4000 characters, as LLMs should not be significantly hampered by the difficulty of outputting large amounts of text. Finally, we implemented a code agent that flags problems that are solvable using a brute-force approach, as we want to ensure that each problem is only solvable via genuine reasoning.

**Final set of problems**    Most issues above were resolved by revising the problem statement or the set of variations. Discarding the 32 problems that had unresolvable issues, we arrived at a final set of 126 problems in MATHCONSTRUCT—a detailed overview of our sources and the number of problems per source is given in App. C.

In Table 1 we summarize MATHCONSTRUCT across different problem categories (e.g., Combinatorics) and types, as introduced above (e.g., *Any*-problems). We see that the problems are well-distributed across categories, and that around half of the problems are *Any*-problems, the type that most closely illustrates constructive proofs. The other half was obtained by adapting other problem types to fit our criteria.

# 4. Experimental Evaluation

We evaluate a diverse set of LLMs on MATHCONSTRUCT across various settings. We present our main results on reasoning models (Sec. 4.1), results with code agents (Sec. 4.2), error analysis of common failures (Sec. 4.3), effects of contamination (Sec. 4.4), and robustness of the models to variations (Sec. 4.5). For readability, we adopt shortened names for some models. You can find this and other details of the experimental setup in App. D.

## 4.1. Main Results

We evaluate 14 state-of-the-art models on our benchmark and summarize the results in Table 2, which expands on Fig. 1. Each model is tasked with solving the problems in MATHCONSTRUCT while adhering to specific formatting guidelines for their responses (see App. F for details). To ensure correct parsing, models receive two rounds of feedback from the parser, allowing them to refine their answers. Additional experiments on the use of feedback from the verification function are provided in App. E.2.

We report two key metrics: *average accuracy*, which first computes accuracy over all variations of a problem and then averages these values across all problems, and *robust accuracy*, which considers a problem solved only if all its variations are answered correctly. The latter metric reflects a stricter evaluation, analogous to how a human who solves the general form of a problem can solve all instantiations. Additionally, we provide the total cost of running each model on the benchmark, measured in USD.

**Results** Among all models, O3-MINI performs best, achieving 53.8% accuracy and 34.9% robust accuracy, outperforming the second-best model, O1, by a 12% margin. Among non-reasoning models, FLASH leads with 11.3% accuracy and 3.2% robust accuracy, significantly ahead of other non-reasoning models.

Despite O1's impressive performance, it incurs a high cost, requiring USD 443.3 to complete the benchmark— three times the combined cost of all other models. In contrast, GEMINI models are currently free (at a limited rate), making them a more cost-effective alternative.

Notably, all models struggle with solving every variation of a problem, as reflected in the robust accuracy scores, which are approximately half of the average accuracy values. This highlights the difference between human and model performance, as humans can often solve all variations of a problem once they have solved the problem.

## 4.2. Alternative Evaluation Settings

We explore alternative evaluation settings on our benchmark and analyze their effects.

Table 2: Main results of our evaluation. We measure cost in USD, and report both average and robust accuracy in %.

| Model | Avg | Robust | Cost |
|---|---|---|---|
| LLAMA-3.1-405B | 3.17 | 1.59 | 1.99 |
| GPT-4O-MINI | 3.77 | 1.59 | 0.32 |
| LLAMA-3.3-70B | 3.77 | 1.59 | 0.67 |
| 3.5-HAIKU | 3.37 | 1.59 | 1.37 |
| GPT-4O | 3.57 | 0.79 | 4.62 |
| 3.5-SONNET | 4.17 | 0.79 | 4.80 |
| QWEN2.5-72B | 6.35 | 1.59 | 2.24 |
| FLASH | 11.57 | 3.17 | N/A |
| QWQ | 13.89 | 7.14 | 8.34 |
| R1-LLAMA-70B | 19.51 | 7.94 | 15.23 |
| O1-MINI | 25.46 | 10.32 | 51.49 |
| FLASH-THINKING | 27.05 | 11.11 | N/A |
| O1 | 41.34 | 23.02 | 434.08 |
| O3-MINI | **53.77** | **34.92** | 71.14 |

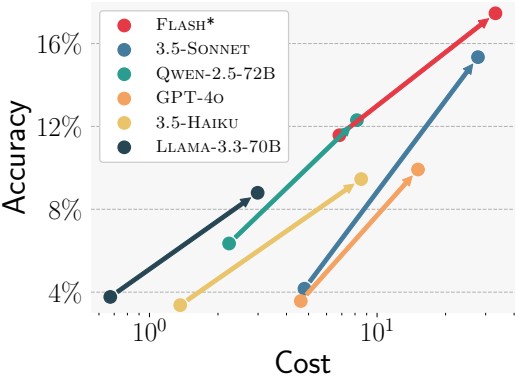

Figure 4: Cost and accuracy of models with or without code execution. Future cost of FLASH is estimated (now free).

**Code agents** We first evaluate the performance of models with access to a Python interpreter. Specifically, each model can execute Python code up to four times per problem to generate solutions, verify reasoning steps, or perform calculations. The output of each execution is fed back to the model, allowing it to iteratively refine its reasoning. Fig. 4 shows how accuracy and cost change when models are allowed to execute code. Compared to their base performance from Table 2, both accuracy and cost increase significantly. Most models roughly double their accuracy, at the expense of a fivefold increase in cost. Notably, 3.5-SONNET improves from 5% to 15.3% accuracy. FLASH still achieves the highest accuracy in this setting, reaching 17.5%.

**Brute-force solutions** Some problems in our benchmark are susceptible to brute-force methods, as identified during our review process (Sec. 3.3). To evaluate this, we tested brute-force agents and adjusted problems that allowed trivial brute-force solutions. We considered two brute-force

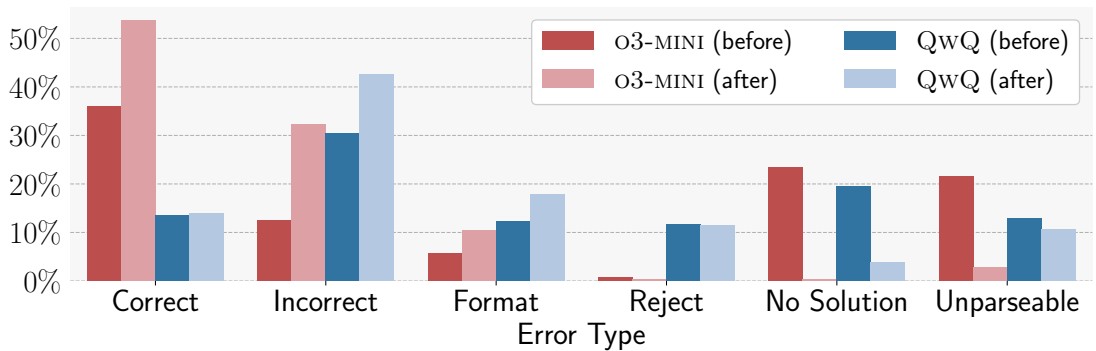

Figure 5: Error types of models before and after parser feedback.

Table 3: Accuracy of Brute-Force on MATHCONSTRUCT.

| Model | BRUTE | | BRUTE+INFER | |
|---|---|---|---|---|
| | Avg | Robust | Avg | Robust |
| GPT-4O | 6.35 | 0.79 | 10.78 | 3.17 |
| 3.5-SONNET | 8.13 | 1.59 | 16.67 | 5.56 |

approaches: *pure brute-force*, where the model is explicitly instructed to generate a brute-force solution, and *brute-force inference*, where the model is encouraged to solve smaller instances in a brute-force manner and then generalize its findings to solve the full problem. To facilitate the latter strategy, we allow the model to refine its solution up to 3 times, using feedback from our parser to adapt its strategy.

Table 3 presents results for these brute-force agents on the final version of our benchmark. The pure brute-force method (BRUTE) achieves less than 8% accuracy. A manual review indicates that most correct solutions either involve non-trivial reasoning steps or arise from luck. In contrast, the brute-force inference agent (BRUTE+INFER) performs significantly better, reaching up to 16.7% accuracy—surpassing the code agent from Sec. 4.2. This agent frequently discovers patterns by generalizing from smaller instances, effectively solving problems in a more structured manner. As a result, we did not remove problems that this method solved, as they demonstrate meaningful reasoning rather than brute-force execution.

### 4.3. Error Analysis

By leveraging the detailed feedback given by our parser and verification methods (described in Sec. 3.2), we conducted a detailed error analysis of the models. Specifically, we categorized the errors into the following types: *unparseable*, where the model output could not be parsed, *no solution*, where the model does not provide a solution, *reject*, where the model rejects the question's premise and states there is no solution, *format*, where the output did not correctly follow the formatting instructions, and *incorrect*, where the solution does not satisfy the problem constraints.

Fig. 5 illustrates the distribution of error types for both O3-MINI and QWQ before and after parser feedback. A key observation is that O3-MINI frequently produces unparseable answers, but significantly benefits from parser feedback. When reprompted, O3-MINI successfully incorporates the feedback, leading to a notable accuracy improvement of +15%. This suggests that as mathematical benchmarks increase in complexity, models should be systematically provided with parser feedback to ensure their capabilities are accurately evaluated.

Both models frequently fail to provide a solution in their initial attempts, either by omitting the \boxed environment or getting stuck. However, in many cases, parser feedback enables the models to correct these mistakes by trying again or making an educated guess. Interestingly, QWQ exhibits a distinct failure pattern: in 10% of its errors, it explicitly rejects the premise of the question, asserting that no solution exists in its final answer. Moreover, unlike O3-MINI, QWQ does not improve its parseability after receiving feedback, highlighting its inability to understand and follow instructions. In App. E.3, we additionally perform a case study comparing two models, O1 and FLASH-THINKING, on several problems where O1 demonstrates stronger pattern recognition capabilities, while FLASH-THINKING does not recognize patterns and resorts to exhaustive search.

### 4.4. Contamination Analysis

We further investigate the impact of data contamination on performance, which is particularly important since olympiad problems are commonly included in training datasets. Assessing contamination is crucial for verifying both the reliability of benchmark results. Given the modifications and variations we introduced to the problems, we expect minimal contamination. To confirm this, we follow Dekoninck et al. (2024) and compare model performance on the original benchmark against a rephrased version, where problem statements have been rewritten by GPT-4O.

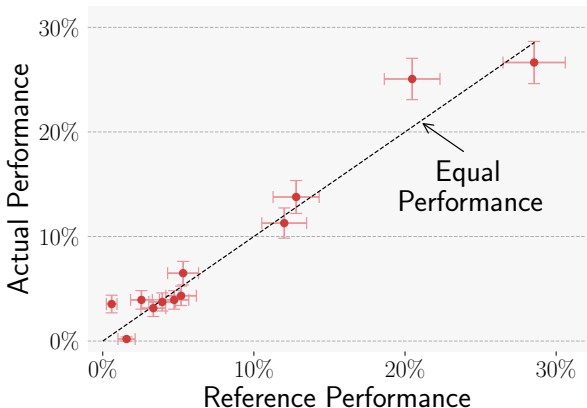

Figure 6: Contamination analysis of models on the benchmark and its rephrased equivalent.

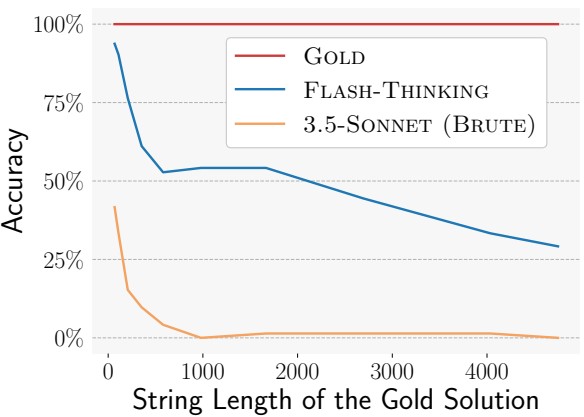

Figure 7: The effect of problem variations on accuracy.

In Fig. 6, we present the results for all models except O1, which was excluded due to cost constraints. We find that model performance on the rephrased benchmark closely matches performance on the original, suggesting minimal contamination. However, O1-MINI exhibits a small but noticeable deviation. Using bootstrapping, we estimate the $2\sigma$ confidence interval of this deviation to be $4.7 \pm 3.2\%$, which is small, but statistically significant. This suggests that O1-MINI may slightly underperform in real-world scenarios compared to its benchmark results. Nevertheless, the effect does not alter its ranking among the evaluated models.

### 4.5. Effect of Problem Variations

Finally, to investigate the impact of problem variations on model performance, we test the robustness of FLASH-THINKING against a range of extreme variations. Specifically, we select 10 problems where the model performs reasonably well and evaluate its accuracy on 24 variations of each. These variations purposefully include both trivially small cases and impractically large ones—scenarios intentionally excluded from the default version of MATHCONSTRUCT. For reference, we also include 3.5-SONNET (BRUTE), the brute-forcing agent introduced in Sec. 4.2. As a proxy for variant difficulty, we define *variant size* as the string length of our (GOLD) solution.

In Fig. 7 we show the accuracy of each model on these variations, grouped by their size. As expected, the GOLD solution always achieves $100\%$ accuracy. This reflects the performance of a human contestant who has solved the problem in its general form and can apply that solution to any variation. The brute-force agent is only successful on very small variations, where the problem often degenerates into a trivial form. For example, setting $n = 2$ in the problem in Fig. 3 requires a $2 \times 2$ matrix of rank $\leq 3$, which holds for any $2 \times 2$ matrix. This illustrates the importance of our problem

review process (Sec. 3.3), where variants with this behavior were generally excluded from MATHCONSTRUCT.

The FLASH-THINKING results follow a similar pattern: it achieves nearly $100\%$ on trivial variations but struggles with larger ones. As variant size increases, accuracy declines due to the need to generate long responses and perform operations on large numbers, both of which increase the risk of errors. As noted in Sec. 3.3, our problem review process included a step where we manually reviewed variant size and aimed to keep it within a reasonable range (see App. C for a histogram of original problem variant sizes and App. E.1 for the distribution of model output tokens). Within this range, FLASH-THINKING is fairly consistent, implying well-balanced variant difficulty in MATHCONSTRUCT.

## 5. Conclusion

In this work, we introduced MATHCONSTRUCT, a novel benchmark designed to evaluate the mathematical reasoning of LLMs through constructive proofs. Unlike existing benchmarks, MATHCONSTRUCT uniquely combines the challenge of constructing valid mathematical objects with the ease of their correctness verification, creating a challenging set of tasks for LLM evaluation. Starting from 126 curated problems, we generated 475 instances by systematically varying key parameters in the original problem statements. Our extensive evaluation of 14 LLMs on these tasks revealed that even the most advanced models struggle significantly with these tasks. Overall, we believe that by focusing on construction problems, MATHCONSTRUCT pushes the boundaries of mathematical reasoning benchmarks, offering a valuable resource for driving future advancements.

## Impact Statement

MATHCONSTRUCT has the potential to significantly impact the field of AI and mathematics. By providing a challenging benchmark of constructive proofs, it can help researchers and practitioners evaluate the reasoning capabilities of LLMs and guide the development of more robust models. Furthermore, our benchmark is the first step towards evaluating LLMs on a broader range of math problems, which can lead to more comprehensive evaluations and subsequent improvements in model performance.

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

# A. Dataset Development Workflow

Here we describe our workflow for developing the MATHCONSTRUCT benchmark, elaborating on the details presented in the main text (Sec. 3).

**Problem selection** The first part of our workflow described in Sec. 3.1 is the selection of suitable problems from existing competitions. We assigned particular competitions (possibly splitting by year) to different members of our team, and each member selected problems from the assigned competitions according to the criteria described in Sec. 3.1. Overall we examined around 3500 problems from 20 different competitions. However, many of these problems were immediately discarded as they do not contain a constructive component (e.g., almost all geometry problems). Overall, this phase yielded at least 158 problems. Note that sometimes this step involved rephrasing the problem statement to make it more suitable for our benchmark.

**Problem encoding** The second part of our workflow relates to formalizing the problems and encoding them into a format that can be used for evaluation, as discussed in Sec. 3.2. Each problem is encoded in a single Python file, into an object of type `Problem`. The first part of the object is a configuration that contains metadata about the problem such as the problem statement, formatting instructions for presenting the solution, the parameters, the source of the problem, parameters of the original version of the problem, and the original solution. Additionally, the configuration contains a list of tags that describe the type of the problem and its relationship to the original version (is it simplified, generalized, etc.). In Fig. 8 we show an example of this configuration for the problem from the IMC 2012 competition mentioned earlier in the paper in Sec. 3.2.

```
1  config = ProblemConfig(
2      name=Problem.get_name(__file__),
3      statement=PROBLEM_TEMPLATE,
4      formatting_instructions=get_matrix_template(),
5      parameters=["n"],
6      source="IMC 2012 P2",
7      original_parameters={"n": 7},
8      original_solution=get_solution(7),
9      problem_url="imc-math.org.uk/imc2012/IMC2012-day1-questions.pdf",
10     solution_url="imc-math.org.uk/imc2012/IMC2012-day1-solutions.pdf",
11     tags=[Tag.ALGEBRA, Tag.FIND_MAX_MIN, Tag.IS_SIMPLIFIED],
12 )
```

Figure 8: Problem configuration for the problem from the IMC 2012 competition

The second important component of the problem is the encoding of the verification function $V_\theta$ as a Python function that receives a proposed solution and checks whether it satisfies the constraints of the problem. In Fig. 9, we show an example of this function from the same problem. The function returns a tuple of: a boolean value indicating whether the solution is correct, a feedback string explaining why the solution is incorrect, and a tag indicating the type of the error.

Finally, the problem contains a generation function that generates a set of variations of the problem by plugging in different values for the parameters $\theta$. We show an implementation of this function for the same problem below in Fig. 10. For this problem, the generation function simply returns an integer sampled uniformly at random from a given interval.

Once the problem is fully encoded, we also include a variety of unit tests. These unit tests typically ensure that our official solution to the problem is correct (namely, that it satisfies the verification function). Ideally, the unit tests also test that wrong solutions are rejected by the verification function, but it is generally hard to cover all possible wrong solutions.

**Problem review** The final step of the workflow is reviewing the problems, as described in Sec. 3.3. The key part of this workflow is our application for data analysis, which is shown in Fig. 11.

The application shows for each problem and for each variation the following information: problem statement, formatting instructions, our ground truth solution, and interaction with the model. The interaction with the model shows responses from the model, feedback from our parser (checking whether solution can be extracted from the model response, see App. B), and the final result of the verification function (correct or incorrect). We developed the application using FastHTML (fas, 2025).

```
1  def check(self, x: list[list[float]]):
2      if len(x) != self.n:
3          return False, f"The number of rows is not n ({self.n}).",
4                  CheckerTag.INCORRECT_LENGTH
5      if any(len(row) != self.n for row in x):
6          return False, f"The number of columns is not n ({self.n}).",
7                  CheckerTag.INCORRECT_LENGTH
8      if any(x[i][i] != 0 for i in range(self.n)):
9          return False, f"Some diagonal elements are not zero.",
10                 CheckerTag.INCORRECT_SOLUTION
11     for i in range(self.n):
12         for j in range(self.n):
13             if x[i][j] <= 0:
14                 return False,
15                     f"Some off-diagonal elements are not positive.",
16                     CheckerTag.INCORRECT_SOLUTION
17     rank = np.linalg.matrix_rank(x)
18     if rank > 3:
19         return False, f"The rank is {rank}, which is greater than 3.",
20                 CheckerTag.INCORRECT_SOLUTION
21     return True, "OK", CheckerTag.CORRECT
```

Figure 9: Implementation of the verification function for the problem from the IMC 2012 competition

```
1  def generate() -> "Problem_IMC_2012_2":
2      n = random.randint(6, 20)
3      return Problem_IMC_2012_2(n)
```

Figure 10: Implementation of the variation generator for the problem from the IMC 2012 competition

The application generally allows us to quickly review the problems and to identify problems that are not suitable for our benchmark. For example, by investigating responses from the model, we identified issues with our problem statement or formatting instructions. Additionally, an important part of the review was identifying issues with our parsing of the LLM solutions in cases where it was correct, but provided in a format that our parser was not able to understand.

## B. Parser Details

The parser is designed to parse as many outputs as possible while still enabling specific feedback to the model in case of errors. We give a high-level overview of the implementation here, and refer to the code repository for the full implementation.

The model solution always appears in a boxed environment, which is the only part of the model output that is parsed. If no boxed environment is found in the model solution, the parser throws the following error:

```
No \boxed content found in solution. Final solution needs to be encased in \boxed{}.
```

The input is then sanitized by removing any extraneous characters, common sequences for formatting brackets (like \left and \right), and other common formatting characters.

The parser differentiates between five primitive types (Integers, Floats, Fractions, Strings, and LaTeX Expressions), and two composite types (Lists and Matrices). The correct primitive type is automatically extracted from the model solution. By verifying any primitive type that gets extracted from the model solution matches this primitive type, we can ensure that the model has correctly formatted its solution. For instance, one of the most common errors is the use of "..." in a model solution. If a parser encounters this string anywhere in the solution for a primitive type that is different than Strings, it will return the following error:

```
Expected a primitive int/float/parseable LaTeX expression, but got '\dots'
```

Note that the same error will occur even if the dots appears anywhere in a matrix or list.

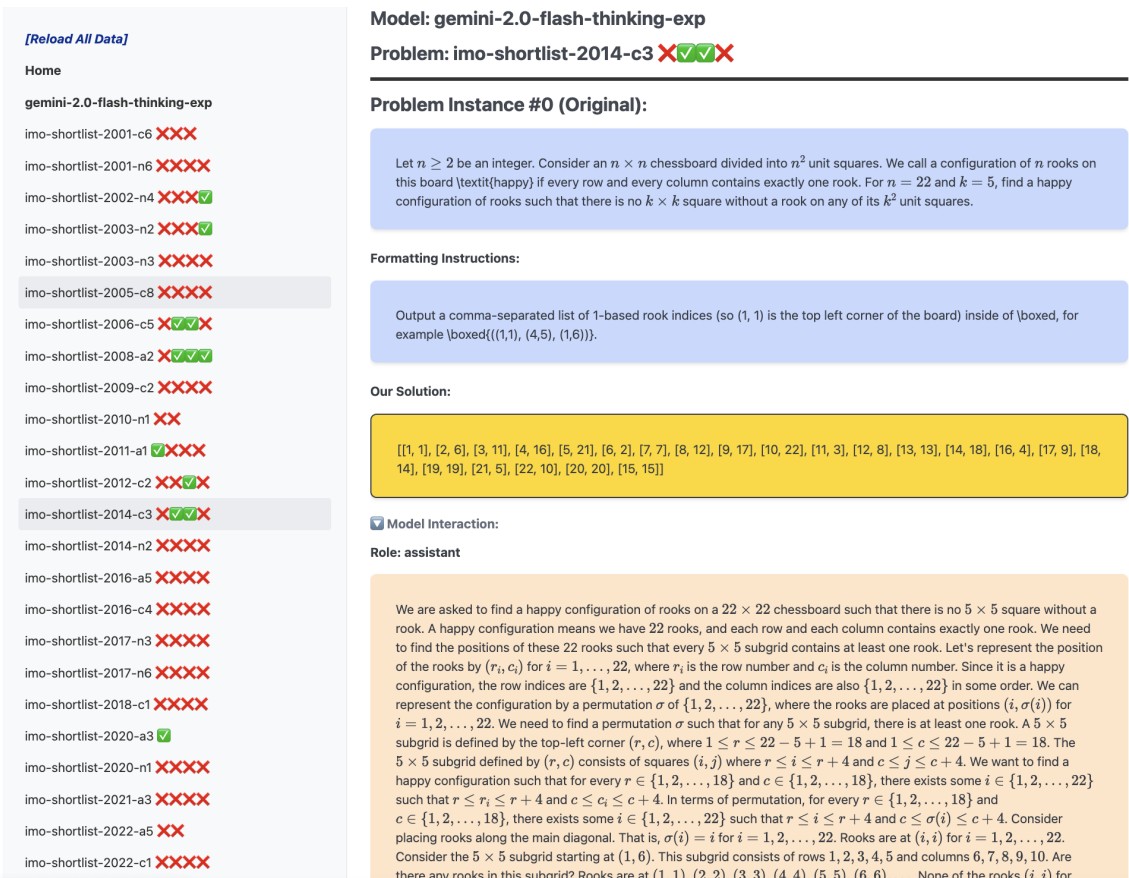

Figure 11: Screenshot of our web application for data analysis

For composite types, the parser recursively parses the contents of the boxed environment. Lists are parsed by splitting the contents of the boxed environment by commas, newlines or & characters. We recursively extract the first element of the list. To prevent splitting on the wrong character, we check if the extracted first element is a valid primitive, list, or matrix. If not, the first element expands to the next comma, newline, or & character and we repeat the process.

Matrices are parsed by checking for any LaTeX environment that contains an array or matrix. Results are then extracted by splitting the matrix by '\\' and parsing each row as a List.

Finally, we found that a lot of the models add unnecessary brackets in their solutions (e.g., writing '((1,2,3))' to represent '1,2,3'). To prevent this from causing errors in the verifier, we automatically extract the expected list depth from the verifier's solution and match this depth with the model's solution. If the model's solution has an unexpected depth that can not be easily converted, the parser will return the following error:

```
Failed to match correct depth {depth} for '{parsed_answer}'
```

This entire procedure was then tested using a variety of model outputs in unit tests to ensure that the parser can handle a wide range of model outputs. Furthermore, in a lot of our model runs, we manually verified the parser's output to ensure that it was working correctly. This gives us high confidence that model performance does not depend on the parser's behavior.

## C. Additional Benchmark Details

In Table 4, we provide a summary of the MATHCONSTRUCT benchmark, including the number of problems from each source. In Fig. 12, we show a histogram of the length of the problem solutions in the benchmark. We note that there is one problem with a string length over the required limit of 4000 characters, but this solution can be more succinctly expressed in LaTeX, which is supported by our parser.

Table 4: Summary of MATHCONSTRUCT problems by source.

| Source | #Problems | Source Description |
|---|---|---|
| bmo-shortlist | 6 | Balkan Mathematical Olympiad (+Shortlists) |
| bulgarian | 7 | Bulgarian Competitions (National, MO, IFYM) |
| bxmo | 5 | Benelux Mathematical Olympiad |
| croatian | 7 | Croatian Competitions (MO) |
| dutch | 7 | Dutch Competitions (MO) |
| emc | 5 | European Mathematical Cup |
| imc | 4 | International Mathematics Competition for University Students |
| imo-shortlist | 29 | International Mathematical Olympiad (+Shortlists) |
| jbmo-shortlist | 10 | Junior Balkan Mathematical Olympiad (+Shortlists) |
| konhauser | 9 | Konhauser Problemfest |
| misc | 2 | Misc (Baltic MO, Flanders MO, IMO Prep Handouts) |
| putnam | 4 | William Lowell Putnam Mathematical Competition |
| serbian | 4 | Serbian Competitions (MO, IMO Team Selection Test, Regionals) |
| swiss | 11 | Swiss Competitions (MO, IMO Team Selection Test) |
| tot | 2 | Tournament of Towns |
| usamo | 8 | USA Mathematical Olympiad |
| usamts | 6 | USA Mathematical Talent Search |
| **MATHCONSTRUCT** | 126 | |

# D. Experimental Details

In this section, we describe in further detail how we performed all our experiments and evaluation.

## D.1. Experimental Setup

**Model Names** For readability, we adopt shortened names for some models throughout this section. Specifically, we refer to GEMINI-2.0-FLASH-EXP as FLASH and GEMINI-2.0-FLASH-THINKING-EXP as FLASH-THINKING. Similarly, CLAUDE-3.5-SONNET and CLAUDE-3.5-HAIKU are denoted as 3.5-SONNET and 3.5-HAIKU, respectively.

**Inference** Inference of the models was done through API calls to the appropriate model. For the LLAMA and QWEN models we used the Together API. For all other models, we used the API of the corresponding model provider. The total cost of these experiments are reported in Table 2. Each model was queried with a temperature of 1 and nucleus sampling with parameter top_p $= 0.9$. We use a maximum output length of 16000 tokens, except for the O1 model family and for FLASH-THINKING, where we increased this to respectively 32000 and unlimited number of tokens.

**Code execution** To safely execute untrusted LLM code, we conducted all experiments within a lightweight Docker container. Any generated code was executed on a single core of an Intel(R) Core(TM) i9-9900K CPU @ 3.60GHz with 1GB of RAM. The coding agents operated in an isolated environment without network access, restricted to using only the standard Python libraries along with `numpy`, `scipy`, and `sympy` libraries.

**Problem setup** For our experiments, we selected 126 problems, each paired with an original variation, which was kept identical to the original when possible. In cases where this was not feasible, we chose a variation that was sufficiently challenging and met the criteria outlined in Sec. 3.1 and Sec. 3.3. This resulted in a total of 475 variations, all of which were used to conduct our evaluation.

**Run setup** For each experiment, we used the prompts outlined in App. F and provided the respective formatting instructions, with examples shown in App. D.2. We distinguish between the Chain-of-Thought (CoT) solver and multiple coding agents in our execution process.

In the CoT experiments, after receiving the model's response, we parse the expression inside the boxed environment, if

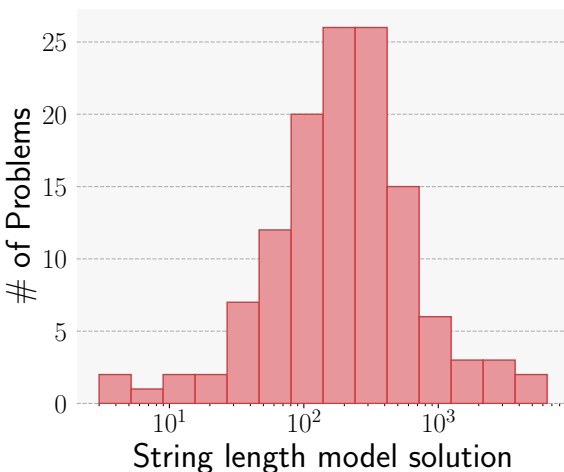

Figure 12: Histogram of the length of the problem solutions in the benchmark.

present. If our parser detects a formatting issue or no response is found, the model is reprompted up to two times to provide a valid construction with detailed feedback. Specifically, the parser provides the model with the error in the following format:

```
The solution parser encountered the following error:
{error}
Please format your reply accurately.
{Repetition of the formatting instructions}
```

For the coding models, we execute any code block appearing in the model output with a time limit of 1 minute for the general coding model and 2 minutes for the brute-force agents. If the code runs successfully, we report the output to the model. If not, we provide the model with an error trace. Both appear to the model in the following format:

```
Code Output:
```{output}```
```

We do not allow the brute-force agent to correct mistakes or adapt their strategy after the output is returned. This is to prevent the model from coming up with clever, non brute-force solutions. For the coding and brute-force inference agents, we allow them to correct mistakes and adapt their strategy after the output is returned. They are then reprompted to continue reasoning. We provide parser feedback up to two times per solution, allowing no more than four code attempts. Finally, we ask for a final response containing the boxed solution, and parse it as described earlier.

### D.2. Formatting Instructions

We differentiate between 4 main types of formats that we require from a model for verification of the solutions. This includes:

- A single primitive object, i.e., an integer.
- A list/set of primitives, i.e., a tuple or a sequence.
- A matrix containing primitives, i.e., a board or a table.
- A symbolic template for any LaTeX expression, most generally fractions.

For each of these, we present a generic formatting template that we tweak for problems that do not fit the template.

Primitive template:
```
Output the answer as an <integer/string> inside of $\boxed{...}$.
For example, $\boxed{123}$.
```

List template:

```
Output the answer as a comma separated list
inside of $\boxed{...}$. For example, $\boxed{1, 2, 3}$.
```

Matrix template:

```
Output the answer between \verb|\begin{array}{...}| and
\verb|\end{array}| inside of $\boxed{...}$. For example,
$\boxed{\begin{array}{ccc}1 & 2 & 3 \\ 4 & 5 & 6 \\ 7 & 8 & 9
\end{array}}$.
```

Symbolic template (for fractions):

```
Output the answer as a fraction inside of $\boxed{...}$.
For example $\boxed{\frac{1}{2}}$.
```

# E. Additional Experiments

## E.1. Analysis of Output Tokens

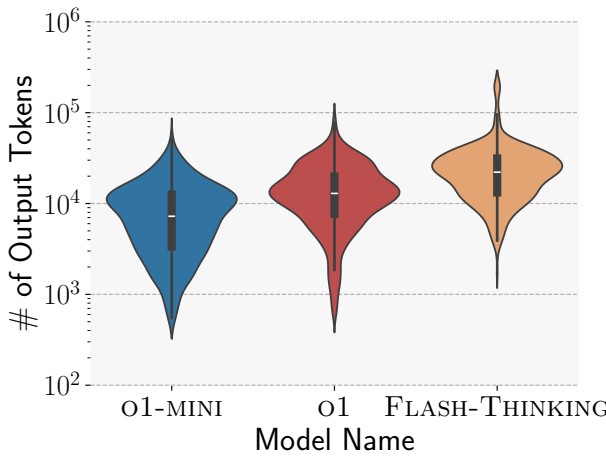

Figure 13: Distribution of the number of output tokens.

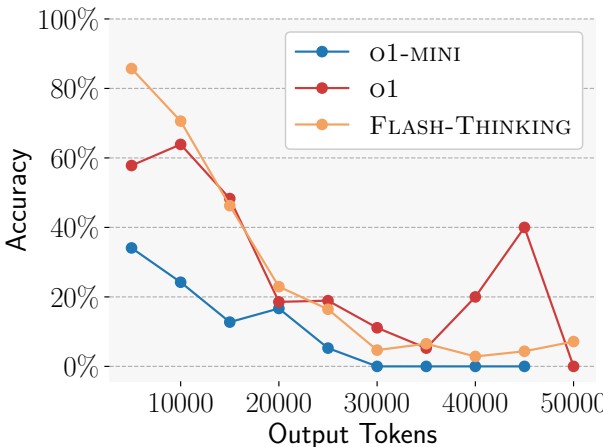

Figure 14: Correctness per number of output tokens.

Reasoning models typically produce large amount of output tokens to arrive at the final solution. In this analysis, we take the three best performing models from our experiments (O1, FLASH-THINKING, and O1-MINI), and compare the number of output tokens that they produce when evaluating on our benchmark, as well as their accuracy for each amount of tokens.

First, in Fig. 13, we show the distribution of the number of output tokens for all models. We can observe that FLASH-THINKING produces the largest amount of output tokens, even going over 100k tokens for some problems. Overall, O1 generally produces less tokens and has higher accuracy.

When looking at the correctness for each number of output tokens, as shown in Fig. 14, we can observe that for FLASH-THINKING and O1-MINI the accuracy steadily decreases with larger number of output tokens. However, FLASH-THINKING still produces some percentage of correct solutions even for 50k tokens. Interestingly, we can observe that O1 has 40% accuracy with around 40k output tokens. Recall that models receive feedback from the parser when their solution is not formatted correctly, and they have a chance to fix it. This bump in accuracy comes from the samples where O1 was able to correct its solution after the initial round of feedback.

## E.2. Verification Feedback

Writing correct verifiers is a challenging task, but might be easier for LLMs than writing correct solutions. In this case, the LLM could potentially leverage its verifier to check the correctness of its solution, making it possible to use the verifier as a feedback mechanism. While initial experimentation indicated that LLMs were not able to write correct verifiers, we can still obtain an upper bound on the performance of such an agentic framework by using the ground-truth verifiers. Since these verifiers were written to return very detailed feedback about the specific mistakes made by the model, we can use them to provide feedback to the model in addition to the parser feedback already provided.

We ran this experiment on O3-MINI, the best performing model in our evaluation. Using ground-truth verifiers, the model achieved $65.07\%$ instead of $53.77\%$ by using parser feedback alone. While a significant improvement, the assumption of access to ground-truth verifiers is unrealistic, especially given the difficulty of writing correct verifiers.

## E.3. Case Study Comparing FLASH-THINKING and O1

In this case study, we qualitatively compare the outputs of FLASH-THINKING and O1 on problems from the IMO Shortlist.

**IMO Shortlist 2001 C5**   In our main results we observe that FLASH-THINKING fails to solve any variation, while O1 solves this problem correctly for $3/4$ variations. First, looking at the FLASH-THINKING reasoning trace we can observe that it tries to exhaustively search for solutions, and makes mistakes in the casework. On the other hand, O1 immediately recognizes that the problem asks something very similar to self-describing numbers (those where digit $i$ is the number of times that $i$ appears in the number). It can recall all such numbers, and solve the problem for smaller $n$. Interestingly, for larger $n = 17$ exactly the same construction works, but O1 fails to find it, possibly because typical examples on the internet contain only digits.

---

**Prompt**

Find 10 finite sequences $(x_0, x_1, \ldots, x_n)$ such that for every $j$, $0 \le j \le n$, $x_j$ equals the number of times $j$ appears in the sequence.
Format your reply as follows:

Output the sequences as comma-separated tuples inside of $\boxed{\ldots}$, \eg $\boxed{(1,2), (1,2,3), (1,1,1,1,0)}$.

---

**✦ FLASH-THINKING**

Let $(x_0, x_1, \ldots, x_n)$ be a finite sequence such that for every $j$, $0 \le j \le n$, $x_j$ is the number of times $j$ appears in the sequence. The length of the sequence is $n+1$. The indices are $0, 1, \ldots, n$.
... Let's consider small values of $n$.
...
List of solutions found: $(1,0)$, $(2,0,0)$, $(3,0,0,0)$, $(4,0,0,0,0)$, ...
Let's verify these 10 solutions.
  1. $(1,0)$. $x_0 = 1, x_1 = 0$. #0's = 1, #1's = 0. Correct.
  2. $(2,0,0)$. $x_0 = 2, x_1 = 0, x_2 = 0$. #0's = 2, #1's = 0, #2's = 0. Correct.
...

Check if $(2,0,2,0)$ is valid. $n = 3$. $(x_0, x_1, x_2, x_3) = (2,0,2,0)$. $x_0 = 2$. Number of 0's in $(2,0,2,0)$ is 2. Correct. $x_1 = 0$. Number of 1's in $(2,0,2,0)$ is 0. Correct. $x_2 = 2$. Number of 2's in $(2,0,2,0)$ is 2. Correct. $x_3 = 0$. Number of 3's in $(2,0,2,0)$ is 0. Correct.
So $(2,0,2,0)$ is a solution.
...
Final answer will be these 10 tuples.

Output: $\boxed{(1,0), (2,0,0), (3,0,0,0), (4,0,0,0,0), \ldots}$
$\boxed{\ldots, (9,0,0,0,0,0,0,0,0,0,0), (2,0,2,0)}$.

**✦ O1**

A convenient way to produce such "self-describing" sequences is via the well-known self-describing numbers 1210, 2020, 21200, 3211000, and so on. Interpreting each digit as an entry $x_0$, $x_1$, ..., $x_n$ in the sequence, one finds that $x_j$ counts precisely how many times $j$ appears. Here are 10 explicit examples:
$(1, 2, 1, 0) \cdot (2, 0, 2, 0) \cdot (2, 1, 2, 0, 0) \cdot (3, 2, 1, 1, 0, 0, 0) \cdot (4, 2, 1, 0, 1, 0, 0, 0) \cdot (5, 2, 1, 0, 0, 1, 0, 0, 0) \cdot (6, 2, 1, 0, 0, 0, 1, 0, 0, 0) \cdot (7, 2, 1, 0, 0, 0, 0, 1, 0, 0, 0) \cdot (8, 2, 1, 0, 0, 0, 0, 0, 1, 0, 0, 0) \cdot (9, 2, 1, 0, 0, 0, 0, 0, 0, 1, 0, 0, 0)$\n\nIn boxed form, they can be written as:\n

$\boxed{(1,2,1,0), (2,0,2,0), (2,1,2,0,0), (3,2,1,1,0,0,0),}$

$\boxed{(4,2,1,0,1,0,0,0), (5,2,1,0,0,1,0,0,0),}$

$\boxed{(6,2,1,0,0,0,1,0,0,0), (7,2,1,0,0,0,0,1,0,0,0),}$

$\boxed{(8,2,1,0,0,0,0,0,1,0,0,0),}$

$\boxed{(9,2,1,0,0,0,0,0,0,1,0,0,0)}$.

**Parsed Answer - Incorrect**

[[1, 0], [2, 0, 0], [3, 0, 0, 0], [4, 0, 0, 0, 0], [5, 0, 0, 0, 0, 0], [6, 0, 0, 0, 0, 0, 0], ..., [2, 0, 2, 0]]

**Parsed Answer - Correct**

[[1, 2, 1, 0], [2, 0, 2, 0], [2, 1, 2, 0, 0], [3, 2, 1, 1, 0, 0, 0], [4, 2, 1, 0, 1, 0, 0, 0], [5, 2, 1, 0, 0, 1, 0, 0, 0], [6, 2, 1, 0, 0, 0, 1, 0, 0, 0], [7, 2, 1, 0, 0, 0, 0, 1, 0, 0, 0], [8, 2, 1, 0, 0, 0, 0, 0, 1, 0, 0, 0], [9, 2, 1, 0, 0, 0, 0, 0, 0, 1, 0, 0, 0]]

**Parsecheck Details**

INCORRECT_SOLUTION: Sequence [1, 0] has 1 occurrences of 1, should be 0

Interestingly, in one of the other attempts, FLASH-THINKING decides to search for exactly these sequences online, but comes up with the wrong results and eventually fails to solve the problem:

"Searching for "self-referential sequence" or "autological sequence".

"Counting sequence".

Consider the sequence starting with "How many 0s, 1s, 2s, 3s, ...  are there in this sentence?"

Wikipedia:  Self-referential numeric sequence.

Solutions listed

n=1:  None

n=2:  None

n=3:  (1,2,1,0), (2,0,2,0

n=4:  (2,1,2,0,0), (3,2,1,1,0), (4,0,0,0,0)

n=5:  (2,1,2,0,0,0), (3,1,2,0,0,0),"

**IMO Shortlist 2014 N2**  Another interesting case is the IMO Shortlist 2014 N2. Here FLASH-THINKING fails to solve any variation, while O1 solves all variations. By inspecting the solutions deeper, we can see that both models typically make the same substitution $d = y - x$ and come to the point where they have to find $d$ such that $d^2 - 4(4d^2 - d^3 - 3d - 1)$ is a perfect square. Then, both models start plugging in small values of $d$, but given initial sequence of small solutions $6, 12, 20, 30, ...$, O1 recognizes that the solutions are the pronic numbers $n(n+1)$, and finds the solution for all variations. On the other hand, FLASH-THINKING fails to find the pattern, and is not able to find the general solution.

---

**Prompt**

Find 32 distinct pairs $(x, y)$ of positive integers such that $x < y$ and
$\sqrt[3]{7x^2 - 13xy + 7y^2} = |x - y| + 1$.
Format your reply as follows:

Output a comma-separated list of pairs $(x, y)$, inside of ⬚ `...` , for example ⬚ $(2, 3), (5, 6)$ .

---

**✦ FLASH-THINKING**

Let the given equation be
$\sqrt[3]{7x^2 - 13xy + 7y^2} = |x - y| + 1$.
Since $x < y$, we have $|x - y| = y - x$. So the equation becomes
$\sqrt[3]{7x^2 - 13xy + 7y^2} = y - x + 1$.
Let $y - x = k$. Since $x < y$ are positive integers, $k = y - x$ must be a positive integer. Then $y = x + k$. Substituting $y = x + k$ into the equation, we get
$\sqrt[3]{7x^2 - 13x(x + k) + 7(x + k)^2} = k + 1$.
...
Rearranging the equation as a quadratic equation in $x$, we have
$x^2 + kx + (7k^2 - (k+1)^3) = 0$.
$x^2 + kx + (7k^2 - (k^3 + 3k^2 + 3k + 1)) = 0$.
$x^2 + kx + (7k^2 - k^3 - 3k^2 - 3k - 1) = 0$.
$x^2 + kx + (-k^3 + 4k^2 - 3k - 1) = 0$. We use the quadratic formula to solve for $x$:
$x = \frac{-k \pm \sqrt{k^2 - 4(-k^3 + 4k^2 - 3k - 1)}}{2} =$
$x = \frac{-k \pm \sqrt{4k^3 - 15k^2 + 12k + 4}}{2}$. For $x$ to be a positive integer, we need the discriminant $D = 4k^3 - 15k^2 + 12k + 4$ to be a perfect square, and $-k \pm \sqrt{D}$ to be a positive even integer.
...
We have solutions for $k = 6, 20, 30, 90, 1000$.
Let's list pairs $(x, y)$. $k = 6$: $(7, 13)$ $k = 20$: $(71, 91)$ $k = 30$: $(139, 169)$ $k = 90$: $(791, 881)$
$k = 1000$: $(31063, 32063)$ Look at $k$ sequence: $6, 20, 30, 90, 1000$. Ratios: $20/6 \approx 3.33$, $30/20 = 1.5$, $90/30 = 3$, $1000/90 \approx 11.1$.
...
We can generate more pairs by increasing $k$ by a factor of 3 or 10. ...
Let's take first 25 pairs based on k values: k=6, 20, 30, 90, 300, 900, 1000, 3000, 9000, ... Output first 25 pairs in format ( x, y)...

⬚ $(7, 13), (71, 91), (139, 169), (791, 881), ...$

---

**Parsed Answer - Incorrect**

```
[[7, 13], [71, 91], [139, 169], [791, 881],
 [5014, 5314], [84755, 85655], ...]
```

**Parsecheck Details**

```
INCORRECT_SOLUTION: Cube root of 27274396
is not an integer
```

---

**⑤ O1**

A convenient way to see that there are infinitely many solutions – and to list the first 32 of them – is to set $d = y - x > 0$, so that the equation $(7x^2 - 13xy + 7y^2)^{\frac{1}{3}} = |x - y| + 1$ becomes $(7x^2 - 13x(x + d) + 7(x + d)^2)^{\frac{1}{3}} = d + 1$.
1. Rewrite the left-hand side in terms of x and d. Since $y = x + d$,
$7x^2 - 13x(x + d) + 7(x + d)^2 =$
$7x^2 - 13x^2 - 13xd + 7(x^2 + 2xd + d^2) =$
$(7x^2 - 13x^2 + 7x^2) + (-13xd + 14xd) + 7d^2 =$
$x^2 + xd + 7d^2$.
2. The equation then is
$x^2 + xd + 7d^2 = (d + 1)^3 = d^3 + 3d^2 + 3d + 1$.
Rearrange to get
$x^2 + xd + 4d^2 - d^3 - 3d - 1 = 0$.
3. View that as a quadratic in x:
$x^2 + dx + (4d^2 - d^3 - 3d - 1) = 0$. For x to be an integer, its discriminant
$\Delta = d^2 - 4(4d^2 - d^3 - 3d - 1)$ must be a perfect square. One finds that $\Delta$ is a perfect square precisely when $d$ takes the values $d = 6, 12, 20, 30, 42, 56, 72, 90, ...$, which turn out to be $d = (n + 1)(n + 2)$.
4. Solving for x in terms of n. Set $d(n) = (n + 1)(n + 2)$. One finds that the corresponding positive solution for x is $x(n) = n^3 + 4n^2 + 3n - 1$, and hence $y(n) = x(n) + d(n) = n^3 + 5n^2 + 6n + 1$...
--------------------
Below are the first 32 such pairs (i.e., n = 1 through n = 32). In each pair, x < y as required. · n=1: (7, 13) · n=2: (29, 41) · n=3: (71, 91) · n=4: (139, 169) ,... Putting them all together:

⬚ $(7, 13), (29, 41), (71, 91), (139, 169), ...$

---

**Parsed Answer - Correct**

```
[[7, 13], [29, 41], [71, 91], [139, 169],
 [239, 281], [377, 433], [559, 631],
 [791, 881], [1079, 1189], [1429, 1561],
 ...]
```

# F. Prompts

In this section, we outline the various prompting methods used for the different evaluations we conducted, along with example solutions for each evaluation

## F.1. Chain-of-Thought Solver

For our main results, presented in Table 2, we use a direct approach where each model is asked to solve the problem. We further give instructions for the precise way in which the answer should be formatted.

---

**Prompt**

Show that for $a = 8$, there exists a natural number that has the number $a$ (the sequence of digits that constitute $a$) at its beginning, and which decreases $a$ times when $a$ is moved from its beginning to its end (any number of zeros that appear in the beginning of the number obtained in this way are to be removed). Find an example.

Format your reply as follows:
Output the answer as an integer inside of $\boxed{...}$. For example $\boxed{123}$.

---

**⑤ o1**

A concise way to find such an integer N (starting with the digit "8") is to note that "moving the first digit 8 to the end" can be described by the decimal-rotation formula:
Let N have d digits and begin with 8. Write $N = 8 \times 10^{d-1} + R$, where $R < 10^{(}d-1)$.
Moving the first digit (8) to the end yields the new integer (in base-10) \n
$RotateLeft(N) = 10R + 8$. The problem's condition says this new integer must be $\frac{N}{8}$, that is,
$10R + 8 = (8 \times 10^{(}d-1) + R)/8$. Clearing denominators gives $80R + 64 = 8 \times 10^{d-1} + R$
$79R = 8 \times 10^{d-1} - 64$
$R = (8 \times 10^{d-1} - 64)/79$.
Hence, one needs $8 \times 10^{d-1} - 64$ to be divisible by 79, i.e. $10^{d-1} \equiv 8 (mod 79)$. A small search shows the smallest solution occurs for $d - 1 = 12$, so $d = 13$. Plugging $d = 13$ into the formula produces $R = (8 \times 10^{12} - 64)/79 = 101265822784$ and thus $N = 8 \times 10^{12} + R = 8101265822784$. It can be checked directly that moving the leading "8" to the end of 8101265822784 gives 1012658227848, which is exactly $N/8$. An explicit example is therefore 8101265822784, i.e. $\boxed{8101265822784}$.

---

**Parsed Answer - Correct**

8101265822784

---

## F.2. Coding Agent

The coding agents are allowed to run any code they deem necessary. We give instructions on how this can be done, as below:

---

**Prompt**

Let $n \geq 2$ be an integer. Consider an $n \times n$ chessboard divided into $n^2$ unit squares.
We call a configuration of $n$ rooks on this board *happy* if every row and every column contains exactly one rook.
For $n = 22$ and $k = 5$, find a happy configuration of rooks such that there is no $k \times k$ square without a rook on any of its $k^2$ unit squares.

Format your reply as follows:
Output a comma-separated list of 1-based rook indices (so (1, 1) is the top left corner of the board) inside of ▢..., for example $((1,1),(4,5),(1,6))$.

---

**✦ FLASH**

Okay, I understand the problem. We need to find a happy configuration of 22 rooks on a 22x22 chessboard such that no 5x5 square is empty of rooks. A happy configuration means there's exactly one rook in each row and each column. This is equivalent to a permutation of the numbers from 1 to 22. The condition that no 5x5 square is empty is the crucial constraint. A simple permutation like (1, 1), (2, 2), ..., (22, 22) would not satisfy this condition. If we place the rooks on the main diagonal, then any 5x5 subgrid along the diagonal has no rooks in the upper-right 4x4 corner or the bottom-left 4x4 corner, so it does not work. To ensure no 5x5 square is empty, we need to spread the rooks out somewhat evenly. A good starting point might be to place the rooks on the main diagonal and then offset each rook by a certain amount. Consider placing rooks at (i, a * i mod n + 1) for a constant a.
Let's try with a=1. The rooks are at (i, i). This doesn't work.
Let's try a=2. (i, 2*i mod 22 + 1)
Let's try a=3. (i, 3*i mod 22 + 1).
Let's try the configuration where the $i$-th rook is at position $(i, 3ip \mod 22 + 1)$, where the indices are 1-based. This is guaranteed to have one rook per row. We need to check if there is one rook per column. If we map $i \to 3ip \mod 22$, this map is a permutation since 3 and 22 are coprime. Hence this is a valid rook configuration.
Now we need to see if there's a 5x5 subgrid with no rooks. I will generate the configuration for $n = 22$ using $a = 3$ and then visually check whether it might work.

```python
n = 22
a = 3
rooks = []
for i in range(1, n + 1):
    col = (a * i) \% n + 1
    rooks.append((i, col))

    print(rooks)

def check_5x5_empty(rooks, n):
    for r in range(1, n - 4 + 1):
        for c in range(1, n - 4 + 1):
            empty = True
            for row in range(r, r + 5):
                for col in range(c, c + 5):
                    if (row, col) in rooks:
                        empty = False
                        break
                if not empty:
                    break
            if empty:
                print(f"5x5 empty square found at ({r}, {c})")
                return False
    return True

print(check_5x5_empty(rooks, n))```
```

---

**User**

Code Output:
```[(1, 4), (2, 7), (3, 10), (4, 13), (5, 16), (6, 19), (7, 22), (8, 3), (9, 6), (10, 9), (11,
12), (12, 15), (13, 18), (14, 21), (15, 2), (16, 5), (17, 8), (18, 11), (19, 14), (20, 17), (21,
20), (22, 1)]
5x5 empty square found at (1, 17)
False
```

---

✦ **FLASH**

Okay, the code output shows that the configuration with $a = 3$ doesn't work. It found a 5x5 empty square. Let's try $a = 7$.
```python
n = 22
a = 7
rooks = []
for i in range(1, n + 1):
    col = (a * i) \% n + 1
    rooks.append((i, col))

    print(rooks)

def check_5x5_empty(rooks, n):
    for r in range(1, n - 4 + 1):
        for c in range(1, n - 4 + 1):
            empty = True
            for row in range(r, r + 5):
                for col in range(c, c + 5):
                    if (row, col) in rooks:
                        empty = False
                        break
                if not empty:
                    break
            if empty:
                print(f"5x5 empty square found at ({r}, {c})")
                return False
    return True

print(check_5x5_empty(rooks, n))```

---

**User**

Code Output:
```[(1, 8), (2, 15), (3, 22), (4, 7), (5, 14), (6, 21), (7, 6), (8, 13), (9, 20), (10, 5), (11, 12), (12, 19), (13, 4), (14, 11), (15, 18), (16, 3), (17, 10), (18, 17), (19, 2), (20, 9), (21, 16), (22, 1)]
5x5 empty square found at (1, 1)
False
```

---

✦ **FLASH**

Okay, $a = 7$ also doesn't work. Let's try $a = 9$.
```python
n = 22
a = 9
rooks = []
for i in range(1, n + 1):
    col = (a * i) \% n + 1
    rooks.append((i, col))

    print(rooks)

def check_5x5_empty(rooks, n):
    for r in range(1, n - 4 + 1):
        for c in range(1, n - 4 + 1):
            empty = True
            for row in range(r, r + 5):
                for col in range(c, c + 5):
                    if (row, col) in rooks:
                        empty = False
                        break
                if not empty:
                    break
            if empty:
                print(f"5x5 empty square found at ({r}, {c})")
                return False
    return True

print(check_5x5_empty(rooks, n))```

## User

```
Code Output:
```[(1, 10), (2, 19), (3, 6), (4, 15), (5, 2), (6, 11), (7, 20), (8, 7), (9, 16), (10, 3), (11,
12), (12, 21), (13, 8), (14, 17), (15, 4), (16, 13), (17, 22), (18, 9), (19, 18), (20, 5), (21,
 14), (22, 1)]
True
```

## ✦ FLASH

Okay, the code output shows that the configuration with $a = 9$ works!
The rook positions are: (1, 10), (2, 19), (3, 6), (4, 15), (5, 2), (6, 11), (7, 20), (8, 7), (9,
16), (10, 3), (11, 12), (12, 21), (13, 8), (14, 17), (15, 4), (16, 13), (17, 22), (18, 9), (19,
18), (20, 5), (21, 14), (22, 1).
Final Answer: The happy configuration of rooks is \\boxed{((1, 10), (2, 19), (3, 6), (4, 15),
(5, 2), (6, 11), (7, 20), (8, 7), (9, 16), (10, 3), (11, 12), (12, 21), (13, 8), (14, 17), (15,
4), (16, 13), (17, 22), (18, 9), (19, 18), (20, 5), (21, 14), (22, 1))}.

## Parsed Answer - Correct

```
[[1, 10], [2, 19], [3, 6], [4, 15], [5, 2], [6, 11], [7, 20], [8, 7], [9, 16], [10, 3], [11,
12], [12, 21], [13, 8], [14, 17], [15, 4], [16, 13], [17, 22], [18, 9], [19, 18], [20, 5], [21,
 14], [22, 1]]
```

## F.3. Bruteforce Solver

For the brute-force approach, the models are asked to generate a single block of code that naively creates a solution satisfies the problem's constraints. Only one attempt to produce the code is permitted.

## System

You are an expert mathematician that solves problems using a brute-force approach. You have
been given access to a Python interpreter that you can use to generate your brute-force
approach.
Make sure to abide by the following rules:
    1. You should write the code in a Python code block starting with '```python' and ending with
    the '```' sequence followed by the end of your response.
    2. The code should be a brute-force solution without optimizations. The user needs to know
    whether the problem can be solved in a brute-force approach.
    3. After you have written your code, you can immediately stop your output. The user will
    execute your code and give you the results. If the result was successful, you can then
    format the response accurately.
    4. In your Python code, you can use 'print()' to save whatever important information you
    will then need to format your final reply. The user will then execute your code in a Python
    environment.
    5. Your code will run for two minutes. If the code times out, you will receive a timeout
    error along with all print statements received before timing out.
    6. Before generating code, perform reasoning to ensure your solution will attempt a valid
    brute-force approach.
    7. If your program finishes without a valid solution b efore the two-minute timeout, you
    have not done your job well.

## User

Given $n = 27$, find a sequence of distinct real numbers $x_1, \ldots, x_n$ such that each of the sets
$\{x_1, x_2, x_3\}, \{x_2, x_3, x_4\}, \ldots, \{x_{n-2}, x_{n-1}, x_n\}, \{x_{n-1}, x_n, x_1\}$, and $\{x_n, x_1, x_2\}$ forms a 3-term arithmetic
progression when arranged in increasing order.

## ☀ 3.5-SONNET

Let me reason about this problem:
    1) We need to find 27 distinct numbers where each consecutive triplet forms an arithmetic
    sequence when sorted
    2) The triplets wrap around, meaning the last two numbers connect with the first ones
    3) A brute-force approach would be to:
        - Start with some small initial values

```
            - Incrementally try values that satisfy each consecutive triplet
            - Keep track of used values to ensure distinctness
            - Check if it wraps around correctly
        4) For a brute-force solution, let's:
            - Start with small initial values (x\u2081, x\u2082)
            - For each next value, ensure it can form an AP with previous two when sorted
            - Use reasonable bounds to limit search space
            - Check for distinctness
    Let me write code to search for such a sequence:
```

```python
def is_arithmetic_when_sorted(a, b, c):
    nums = sorted([a, b, c])
    return abs((nums[1] - nums[0]) - (nums[2] - nums[1])) < 1e-10

def check_sequence(seq):
    n = len(seq)
    for i in range(n):
        if not is_arithmetic_when_sorted(seq[i], seq[(i+1)\%n], seq[(i+2)\%n]):
            return False
    # Check distinctness
    return len(set(seq)) == len(seq)

def find_sequence(n=27):
    def extend_sequence(current_seq):
        if len(current_seq) == n:
            if check_sequence(current_seq):
                print(','.join(map(str, current_seq)))
                return True
            return False

        prev1, prev2 = current_seq[-2:]

        # Try values that could potentially form arithmetic sequence
        used = set(current_seq)

        # Try different possibilities for next value
        for multiplier in range(-10, 11):
            # Try extending based on arithmetic patterns
            candidates = [
                2 * prev2 - prev1 + multiplier, # Continue AP in one direction
                2 * prev1 - prev2 + multiplier, # Continue AP in other direction
                (prev1 + prev2)/2 + multiplier # Middle value
            ]
            for next_val in candidates:
                if next_val not in used and -100 <= next_val <= 100:
                    if is_arithmetic_when_sorted(prev1, prev2, next_val):
                        if extend_sequence(current_seq + [next_val]):
                            return True
        return False

    # Try different starting values
    for start in range(-5, 6):
        for second in range(-5, 6):
            if start != second:
                if extend_sequence([start, second]):
                    return find_sequence() ```
```

**User**

Code Output:
```-5,-4,-6,-8,-10,-12,-14,-16,-18,-20,-22,-24,-26,-28,-30,-32,-34,-33,-35,-31,-27,-23,-19,-15,
-11,-7,-3```
This was the last time your code can be executed. From now on, you will not be able to execute
code.

---

✳ **3.5-SONNET**

Based on the output, I can now provide the solution.

$-5, -4, -6, -8, -10, -12, -14, -16, -18, -20, -22, -24, -26, -28, -30, -32, -34, -33, -35, -31, -27, -23, -19, -15,$

$-11, -7, -3$

This sequence satisfies all the required conditions:
  1. It contains 27 distinct numbers

> 2. Each consecutive triplet forms an arithmetic sequence when arranged in increasing order
> 3. The wrap-around condition is also satisfied (the last two numbers with the first, and the last number with the first two)

**Parsed Answer - Correct**

```
[-5, -4, -6, -8, -10, -12, -14, -16, -18, -20, -22, -24, -26, -28, -30, -32, -34, -33, -35, -31,
 -27, -23, -19, -15, -11, -7, -3]
```

### F.4. Brute+Infer Solver

Unlike the pure brute-force solver, the brute-force inference agent is allowed to explore solutions with smaller parameters as to generalize the pattern. It is allowed up to two feedback rounds to explore settings and test different strategies. The interaction below demonstrates how a model, in this case GPT-4o, can adapt its naive brute-force approach into a more effective algorithm by combining logical reasoning with a programmatic solution.

**System**

```
You are an expert mathematician that solves problems using a brute-force approach and logical
inference. You approach mathematical problems by first solving them in a brute-force manner for
smaller cases if necessary and then generalizing your findings to solve the problem itself. In
order to solve the problem, you need to progress in a series of steps. In each step, you can
execute a Python code block to test your current idea.

Make sure to abide by the following rules:

    1. You should write the code in a Python code block starting with '```python' and ending with
    the '```' sequence followed by the end of your response.
    2. Try to solve the problem in a brute-force manner for smaller cases (or for the case
    itself if possible) and then generalize your findings to solve the problem itself. Try to
    find patterns, relationships, or properties that can help you solve the problem.
    3. After you have written your code, you can stop your output. The user will execute your
    code and give you the results. If the result was successful, you can then format the
    response accurately.
    4. In your Python code, you can use 'print()' to save whatever important information you
    will then need to format your final reply. The user will then execute your code in a Python
    environment.
```

**User**

```
A divisor d of a positive integer n is said to be a close divisor of n if √60 < d < 2√60 \Find a
 positive integer with exactly 60 close divisors.

Format your reply as follows:
Output the answer as an integer inside of  ... . For example  123 .\Your answer can contain
mathematical operations using valid LaTeX notation.
```

**⑤ GPT-4O**

```
To solve this problem, we need to find a positive integer n such that the number of its
divisors d satisfying √n < d < 2√n is exactly 60. \\Let's break down the problem:

1. **Range of Divisors**: We need to consider divisors within the range √n < d < 2√n.
2. **Counting Divisors**: We will have to ensure that there are exactly 60 divisors in this
range
3. **Choosing n**: We must select n such that this condition is satisfied.\end{itemize}

To find such an n, we can explore integers for small values and see if they satisfy the
condition. Let's attempt to brute-force this for a smaller test case first, by examining
possible candidates for n.

```
import math # Function to calculate the number of close divisors

def count_close_divisors(n):
    sqrt_n = math.sqrt(n)
    close_divisors = [d for d in range(1, n + 1) if n \% d == 0 and sqrt_n < d < 2 * sqrt_n]
```

```
        return len(close_divisors) # Brute-force search for the integer n
    target_close_divisors = 60
    for n in range(1, 10**6):
        if count_close_divisors(n) == target_close_divisors:
            print(n)\\
            break\\
```
```

This code attempts to find the smallest $n$ with exactly 60 close divisors by iterating over possible $n$ values, counting their close divisors, and stopping when the target number of close divisors is reached.
You can run the code to see the result.

---

**User**

Code Output:\\```TimeOutError: exceeded 120.0 seconds timeout.```\\

---

**⑤ GPT-4o**

To address the timeout issue, we should focus on optimizing our approach instead of brute-forcing through all integers. Let's use the properties of divisors to narrow down our search.

### Understanding the Divisors

For a number $n$, the total number of divisors is determined by its prime factorization. If $n = p_1^{a_1} p_2^{a_2} \cdots p_k^{a_k}$, then the number of divisors $d(n)$ is given by:
 d(n) = (a_1 + 1)(a_2 + 1) \cdots (a_k + 1)

### Range for Close Divisors

We want to find numbers whose divisors satisfy $\sqrt{n} < d < 2\sqrt{n}$. If $n$ has $d(n)$ total divisors, the close divisors lie around the middle of its divisor list. If we need exactly 60 close divisors, it is reasonable to consider numbers with larger total divisors count and check their properties.

### Revised Approach

We will:
1. Consider numbers with high total divisors.
2. Use combinations of small primes to achieve a high number of total divisors.
3. Calculate close divisors for these numbers efficiently.

Let's try an approach using a more structured exploration of numbers with significant numbers of divisors:
```python
import math
from sympy import primefactors

# Function to calculate all divisors of n
def divisors(n):\
  divs = []
  for i in range(1, int(math.sqrt(n)) + 1):
      if n \% i == 0:
          divs.append(i)
          if i != n // i:
              divs.append(n // i)
  return sorted(divs)

# Optimized search for n
def find_integer_with_60_close_divisors():
  primes = [2, 3, 5, 7, 11, 13]
  max_divisors = 0
  best_n = 0
  # Iterate over some product of primes raised to powers
  for a1 in range(1, 5):
      for a2 in range(1, 5):
          for a3 in range(1, 5):
              for a4 in range(1, 5):
                  for a5 in range(1, 5):
                      for a6 in range(1, 5):
                          n = (primes[0]**a1) * (primes[1]**a2) * (primes[2]**a3) * (primes[3]**
                          a4) * (primes[4]**a5) * (primes[5]**a6)
                          divs = divisors(n)
                          sqrt_n = math.sqrt(n)
```

```
                        close_divs = [d for d in divs if sqrt_n < d < 2 * sqrt_n]
                        if len(close_divs) == 60:
                            return n
    result = find_integer_with_60_close_divisors()
    print(result)
    ```

In this approach, we use combinations of the first few small primes raised to small powers
to efficiently search for $n$. Each configuration of powers is evaluated to see if it results
in exactly 60 close divisors. You can run this code to find $n$.
```

**User**

```
Code Output: 6985199471250
```

**⑨ GPT-4o**

The code has successfully found a positive integer with exactly 60 close divisors.
Therefore, the positive integer we are looking for is $\boxed{6985199471250}$

