# OpenReview forum: "MathConstruct: Challenging LLM Reasoning with Constructive Proofs"
_ICML.cc/2025/Conference — ICML 2025 poster_

### Official Review · Reviewer_6y3x · 2025-02-22

**Overall Recommendation:** 4

**Summary:**

This paper introduces MathConstruct, a novel mathematical benchmark designed to evaluate LLMs' reasoning in constructive proofs from high-school competition problems. The authors curated a dataset of 127 challenging problems from various sources and converted them into a unified format with symbolic parameters, enabling parameter variations and generating a total of 480 problem variants.

To rigorously assess LLMs on these tasks, MathConstruct provides Python-based verification code for each problem and incorporates a parser that prompts LLMs and delivers feedback to refine their responses toward the correct output.

Experimental results reveal that current state-of-the-art LLMs (e.g., OpenAI o1) achieve only 41.08% average accuracy on the full set of 480 problems and 22.83% robust accuracy across all variants of the original 127 problems. Additionally, the authors conduct a comprehensive evaluation, employing code agents, brute-force methods, error analysis, and potential data contamination assessments.

**Claims And Evidence:**

From my perspective, all claims made in the submission are supported by clear evidence.

**Essential References Not Discussed:**

I find the related work section to be thorough and well-researched.

**Experimental Designs Or Analyses:**

I think the experiments are comprehensive and sound.

**Methods And Evaluation Criteria:**

I think the proposed benchmark and evaluation setting are well-motivated and make sense.

**Other Comments Or Suggestions:**

I would say the paper is overall very strong, but there are some minor aspects that could be improved. If I understand correctly, the phrase "symbolic problem statement in natural language" in lines 182 and 211 could be more clearly written as "problem statement in natural language with symbolic parameters." Additionally, the authors could consider evaluating DeepSeek R1 on their proposed benchmark to further enhance the comprehensiveness of the evaluation.

**Other Strengths And Weaknesses:**

I find the paper is well-motivated, novel, and overall well-written. It primarily evaluates LLMs on informal constructive proofs, while leveraging Python-based verification methods instead of formal theorem provers, making the approach more lightweight and accessible. The experiments and analysis are also quite comprehensive.

I do not see any major weaknesses in the paper. However, one potential limitation is that constructive proofs in Olympiad mathematics may be relatively rare compared to other types of proof-based or computational problems.

**Questions For Authors:**

In principle, given the original 127 problems in MathConstruct, is it possible to generate an arbitrary number of variants for each problem? What design choices guided the selection of the 480 problem variants used in the benchmark?

**Relation To Broader Scientific Literature:**

I think the proposed benchmark serves as a valuable complement to existing mathematical reasoning benchmarks.

**Theoretical Claims:**

N/A

---

> ### Author Rebuttal · Authors · 2025-03-31
>
> We thank reviewer 6y3x for their review. We are delighted that they recognize MathConstruct as a valuable and novel contribution. We also appreciate their feedback on the clarity of our paper and will incorporate the suggested clarifications. Below, we address their additional questions:
>
> **Q.1. Could the authors evaluate more recent models on MathConstruct?**
> We have evaluated newly released frontier models, including o3-mini and DeepSeek R1, and provide an updated results table below:
>
> | Model | Avg | Robust | Cost |
> |----------------------|--------|--------|-------- |
> | Llama-3.1-405B | 3.17 | 1.59 | 1.99 |
> | GPT-4o-mini | 3.77 | 1.59 | 0.32 |
> | Llama-3.3-70B | 3.77 | 1.59 | 0.67 |
> | 3.5-Haiku | 3.37 | 1.59 | 1.37 |
> | GPT-4o | 3.57 | 0.79 | 4.62 |
> | 3.5-Sonnet | 4.17 | 0.79 | 4.80 |
> | Qwen2.5-72B | 6.35 | 1.59 | 2.24 |
> | Flash | 11.57 | 3.17 | N/A |
> | QwQ-Preview | 13.89 | 7.14 | 8.34 |
> | o1-mini | 25.46 | 10.32 | 51.49 |
> | Flash-Thinking | 27.05 | 11.11 | N/A |
> | R1 | 32.28 | 15.08 | 48.39 |
> | o1 | 41.34 | 23.02 | 434.08 |
> | o3-mini | **53.77** | **34.92** | 71.14 |
>
> We observe that o3-mini significantly outperforms even o1, demonstrating stronger generalization and mathematical reasoning while being considerably more cost-efficient. R1, on the other hand, achieved an accuracy of only 32.3%.
>
> **Q.2. How many variants can be generated, and how were the 480 presented variants selected?**
> Many of our problems, particularly those in the “Find Inf”-category, permit infinitely many variations. However, generating interesting variations is more challenging for other types of problems, as their constructions fail for most values. Additionally, for each problem, we manually defined a range of parameters to ensure two important conditions: (1) the problem is resistant to brute-force solutions, and (2) the resulting output remains within a 4,000-character limit. Most problems allow for at least four variations under these constraints, although a small number permit only two or three. Therefore, we generated up to four variations per problem or fewer if four were not feasible. In practice, many problems could include significantly more variations. This point is illustrated clearly in Figure 7, in which we use 24 variations for each of the ten “Find Inf”-problems discussed in this subsection.

---

> > ### Comment · Reviewer_6y3x · 2025-04-04
> >
> > Thank you for the clarification. I continue to hold a positive assessment of the paper.

---

### Official Review · Reviewer_S8Ey · 2025-03-10

**Overall Recommendation:** 3

**Summary:**

The paper presents MathConstruct, a new benchmark to test LLMs on constructive mathematical proofs with a symbolic verifier for correctness of each problem. Unlike traditional math benchmarks that focus on fixed numerical answer problems,  MathConstruct introduces 127 modified olympiad-level math problems, where tasks require constructing mathematical objects with some specific properties. The also evaluates 13 SOTA LLMs on the dataset, and the best LLM only achieved 41% accuracy, which highlights its difficulty.

**Claims And Evidence:**

The paper claims that MathConstruct is a challenging benchmark for evaluating LLMs in mathematical reasoning.
The evidence is demonstrated by extensive experimental results on various models like GPT4o o1, and compared with brute-force approaches and analysed errors.

**Essential References Not Discussed:**

This paper cites all essential references in this research area, but could add the following:
(i). Program-assisted LLM (PAL) paper in solution enumeration: [1] Gao, Luyu, et al. "Pal: Program-aided language models." International Conference on Machine Learning. PMLR, 2023.
(ii). Datasets: [2] Li, Jia, et al. "Numinamath: The largest public dataset in ai4maths with 860k pairs of competition math problems and solutions." Hugging Face repository 13 (2024).

**Experimental Designs Or Analyses:**

The experimental design seems comprehensive, covering different LLMs and evaluation settings. The error analysis is detailed, revealing common failure modes in LLM.

**Methods And Evaluation Criteria:**

(i) The methods for problem selection, symbolic encoding, and evaluation are well-defined. The benchmark is designed with strict verification criteria, ensuring that models must generate correct constructions rather than relying on memorization.

(ii) Evaluation metrics like average accuracy and robust accuracy in this paper are the correctness of constructed instances on selected variations of problems. However, many problems in this benchmark (e.g. IMO Shortlist 2014 C3 in Appendix B on Page 13) have infinitely many variants (here, variants include \{(n,k) | n, k \in \mathbb{N}^+, n\geq 2, k\geq 1, k \leq \lfloor \sqrt{n-1}\rfloor \}, but evaluations are done on small subsets of the infinite variants, which are insufficient to solve the problem. Overall, the metrics are not suitable enough for evaluating the correctness of problem-solving.

**Other Comments Or Suggestions:**

N/A

**Other Strengths And Weaknesses:**

Strengths:
The paper introduces an interesting and challenging benchmark for evaluating LLM reasoning by constructive math problems.

Weaknesses:
1. The benchmark dataset is very small in size, containing only 127 problems in total. This amount of problems is insufficient to fine-tune an LLM on this task or conduct thorough/in-depth evaluation of constructive math problems.
2. In addition, the dataset curation relies too much on manual laboring (including problem selection, verifier construction, and quality checks) , making it intractable to scale up. More data synthetic methods can be applied to create large-scale high-quality dataset.
3. The “problem variations” in this paper are still limited to specific cases of problems. For example, on Page 4 Figure 3, the “variation” is different cases of $n$ in a problem, instead of a completely new problem adapted from original problem.
4. Regarding the limitations in evaluation metrics, check the Methods and Evaluation Criteria section.
5. The paper mentions that constructive proofs are challenging but does not propose any method to enhance or improve LLM’s capabilities in such tasks. It would be very interesting and contributive if the authors could implement any method and outperform the LLM baselines

**Questions For Authors:**

How do you mitigate potential dataset contamination, especially since olympiad problems are commonly included in pre-training data? Have you provided another set of problems with timestamps after knowledge cutoffs of popular LLMs?

**Relation To Broader Scientific Literature:**

The paper is closely related to mathematical reasoning and somehow contributes to formal theorem proving.

**Theoretical Claims:**

The paper does not make any new theoretical claim.

---

> ### Author Rebuttal · Authors · 2025-03-31
>
> We thank reviewer S8Ey for their review. We appreciate that they found MathConstruct to be an interesting benchmark, and our experimental design detailed. Below, we address their other questions:
>
> **Q.1. Does robust accuracy suitably measure the correctness of problem solving for a given question?**
> Yes, because all included variations require the same underlying solution strategy. Furthermore, we explicitly exclude brute-forceable variations, thereby preventing models from using shortcuts specific to any single problem instantiation. Therefore, if a model correctly applies the solution strategy across all four provided problem variations, it is very likely capable of generalizing this approach to other instantiations.
>
> More fundamentally, the main goal of our benchmark does not depend on these specific metrics. Specifically, our benchmark aims to provide effective means of evaluating models on a new reasoning task, where relative accuracy between models is the primary metric of interest.
>
> **Q.2 Is the small dataset size a limitation?**
> No, this is not a limitation. Although our benchmark is smaller than datasets like MATH [1], it is significantly more challenging, is better verified, and incorporates variations designed to test model generalization. It is worth noting that many widely used benchmarks contain relatively small problem sets. For instance, the 2024/2025 AIME competitions, which are frequently used to evaluate reasoning capabilities, consist of only 30 problems each. Lastly, rigorous benchmarking requires carefully curated problems that are neither erroneous nor overly simplistic. Thus, the importance of high-quality examples outweighs dataset size.
>
> **Q.3. Are there ways to reduce the amount of manual labour involved in the construction of the benchmark?**
> We refer the reviewer to our response to **Q.1** of reviewer Aczy.
>
> **Q.4. What value do the variations add to the benchmark?**
> The variations we define have three main purposes:
> - **Reducing Variance**: Including variations effectively increases the benchmark size, thus reducing measurement variance.
> - **Problem Analysis**: We can perform several additional experiments, such as the one presented in Section 4.5, that analyze the effect of model variations on the model solutions.
> - **Reducing Contamination**: Variations guard against memorization, as a model encountering a known problem from its training data must successfully solve all variations.
>
> **Q.5. Is there a way to improve the performance of LLMs?**
> Yes, we already included several improvements designed to enhance model accuracy. Specifically, we implement parser feedback to ensure accurate interpretation of answers and evaluate agent-based approaches. Furthermore, prompted by the reviewer, we ran additional experiments evaluating an agent-based approach that obtains detailed feedback from the ground-truth verification function. These functions log each failure with a specific reason, and therefore allow the model to correct its answer based on this feedback. Using this approach, o3-mini (the best model) improves by an additional 12% (53%->65%). We stress that this experiment should not be considered valid for real-world applications due to lack of access to the verification function at inference time, but it does give an upper bound on what is possible. We will add a discussion about this result in the appendix of our paper.
>
> We did consider an advanced agent-based approach where the model would first implement this verifier function itself and subsequently use it to validate its solution. However, preliminary experiments checking the automation feasibility of our benchmark showed that current models struggle significantly with this task. Thus, we currently consider this out of scope for this benchmark paper.
>
> **Q.6. Did the authors apply mitigations against contamination?**
> First, we emphasize that our benchmark's problems rarely appear verbatim in existing training datasets. Notably, problems beyond the “Find Any”-category underwent substantial revision to become suitable constructive problems, and some were translated from other languages.
>
> Second, we implemented a contamination detection strategy described in Section 4.4. Previous work shows that rewording mathematical problems can influence model performance in the presence of contamination [2]. By doing so, we observed minimal contamination of our benchmark.
>
> **Q.7. Can you include references to these additional works?**
> We appreciate the reviewer’s suggestion and will incorporate the additional references.
>
> [1] https://arxiv.org/abs/2103.03874
> [2] https://arxiv.org/abs/2405.16281

---

> > ### Comment · Reviewer_S8Ey · 2025-04-09
> >
> > Thanks for authors' reply. While the dataset is small and the proposed method does not fully prove the constructive problems in olympiad math, this paper is still interesting and does introduce a promising path towards informally/formally solving constructive problems. I have raised my rating to 3.

---

### Official Review · Reviewer_Aczy · 2025-03-13

**Overall Recommendation:** 4

**Summary:**

This paper proposes MathConstruct, a new benchmark for mathematical reasoning based on constructive problems from math competitions. These problems are highly interesting, yet benchmarks generally avoid them due to the non unique answers. This paper contributes a suite of problems taken from past along with answer verifiers. The problems are shown to be generally very hard for most LLMs, and OpenAI o1 does substantially better, although also at much higher cost. The authors also propose systematic problem variations to probe for robustness, and evaluate agents with access to a Python interpreter. Most of the problems are hard to brute-force via code alone, although access to the interpreter generally enables models to do better (again with higher cost).

**Claims And Evidence:**

Yes, they are clear

**Essential References Not Discussed:**

N/A

**Experimental Designs Or Analyses:**

Yes. The benchmark construction seems sounds (it involved a significant amount of manual analysis from students with experience in math competitions, so that part is hard to check, yet I think it is inevitable for constructing a high-quality dataset for this domain). The experimental analysis of frontier LLMs also seems sound: I checked the examples in the appendix, and the authors spent a decent amount of effort into getting parsing right, which is often a pain in these evaluations. I believe that the results reflect the model's capabilities, not any unsoundness in the pipeline (e.g. failure to parse answers, etc).

**Methods And Evaluation Criteria:**

Yes

**Other Comments Or Suggestions:**

N/A

**Other Strengths And Weaknesses:**

### Other strengths

The paper finds that performance on standard benchmarks, which would make it seem like competition problems have almost been saturated already (e.g. MATH; and more recent results on AIME by various frontier labs), is not telling the full picture. Even neglected problems from the same difficulty region are still very challenging. The difference in performance between GPT-4o and Sonnet on AIME vs on MathConstruct is quite stark. The paper is clear, the results are significant for the community at large, especially now that mathematical reasoning became one of the main evaluations of frontier LLMs, as advertised extensively by model creators whenever they release models.

### Weakness

The only issue might be that the dataset construction pipeline seems quite effortful for further work to expand. Human-curated datasets are very rarely expanded later (e.g., GSM8k -> GSM1k being a rare example). This doesn't diminish the value of the current benchmark or the results, but I think the work would be more "future proof" if some of the more labor intensive parts could have been automated (e.g., I could imagine an LLM-based pipeline doing a first pass on (a) filtering relevant problems from recent competitions, (b) writing a verifier and ensuring that the example solution verifies, etc, perhaps with manual review by humans only at the end).

**Questions For Authors:**

Did the authors consider automating some parts of the pipeline, as suggested above (besides the quality checks, but I mean more for the construction of the problems per se)? If so, were there significant challenges that prevented this from making into the paper? This would be good for future work to be aware of.

**Relation To Broader Scientific Literature:**

This paper augments the literature in the evaluation of mathematical reasoning in LLMs by providing a class of problems that has been mostly ignored so far, due to the complexity of verifying answers. Although there are many benchmarks in general for "mathematical reasoning", none of them (as far as I know) contains constructive problems.

**Theoretical Claims:**

N/A

---

> ### Author Rebuttal · Authors · 2025-03-31
>
> We thank the reviewer for the constructive comments, and are happy to read that they find our work novel and relevant to the community, and our framework easy to work with. We address any remaining concerns below.
>
> **Q.1. Can any parts of the pipeline be automated? What challenges did the authors encounter when trying to automate them?**
> Although certain elements of our pipeline can be readily automated, most aspects currently remain beyond the capabilities of state-of-the-art agents and models. We detail the three main components of the benchmark curation process below:
> - **Problem Curation**: One aspect of the pipeline where automation was partially successful involves the curation and filtering of constructive problems. Constructive problems can be identified according to the selection criteria specified in our paper. A small portion of our benchmark questions were automatically selected by GPT-4o-mini from a larger internal dataset, and subsequently processed manually by the authors. While there were still plenty of false positives among the selected problems, this did make the selection procedure easier for these questions.
> - **Verifier Function**: The most challenging part to automate is the creation of verification functions. Although we experimented briefly with automating this task, current models were unable to generate rigorous enough verification functions. Developing a robust verification function requires the identification of potential failure modes in model outputs, careful creation of efficient checkers, and creation of thorough test cases that cover all relevant edge cases. While future LLMs might be able to automate this, this currently remains out of reach for complex questions of MathConstruct.
> - **Problem Validation**: Another critical aspect is ensuring problem quality, which falls between problem curation and verifier function generation in terms of automation feasibility. We automated several aspects of this step. For example, confirming problems could not be solved through brute-force methods using LLMs, as detailed in the paper. However, each problem underwent additional validation through an internal peer-review process among the authors. Manual peer review is essential for catching small issues and ensuring nothing is missed. While automation can help, it currently cannot fully replace this process as one needs to be certain that problems are correct once processed through this review. Therefore, even in a fully automated pipeline, some form of human validation would remain necessary.
>
> In conclusion, while investigating further automation opportunities is valuable future work, we emphasize the importance of prioritizing a smaller yet more rigorously curated benchmark to accurately measure model performance.

---

### Decision · Program_Chairs · 2025-05-01

**Decision:**

Accept (poster)

**Comment:**

This research presents a challenging benchmark for assessing the reasoning capabilities of Large Language Models, utilizing constructive math problems and lightweight verifier. The reviewers unanimously provided positive evaluations, commending the novelty of the work, the clarity of its presentation, and the significance of the evaluation outcomes. However, several concerns were raised. These include the limited size of the dataset, the difficulty in automating the pipeline, the inadequacy of the metrics in accurately gauging the correctness of problem-solving approaches, and the relative scarcity of constructive proofs in mathematics. Despite these concerns, AC concurs that this benchmark is a valuable and timely contribution for evaluating reasoning models, and thus recommends acceptance.